# FLAT POSTERIOR DOES MATTER FOR BAYESIAN MODEL AVERAGING

## ABSTRACT

Bayesian neural network (BNN) approximates the posterior distribution of model parameters and utilizes the posterior for prediction via Bayesian Model Averaging (BMA). The quality of the posterior approximation is critical for achieving accurate and robust predictions. It is known that flatness in the loss landscape is strongly associated with generalization performance, and it necessitates consideration to improve the quality of the posterior approximation. In this work, we empirically demonstrate that BNNs often struggle to capture the flatness. Moreover, we provide both experimental and theoretical evidence showing that BMA can be ineffective without ensuring flatness. To address this, we propose Sharpness-Aware Bayesian Model Averaging (SA-BMA), a novel optimizer that seeks flat posteriors by calculating divergence in the parameter space. SA-BMA aligns with the intrinsic nature of BNN and the generalized version of existing sharpness-aware optimizers for DNN. In addition, we suggest a Bayesian Transfer Learning scheme to efficiently leverage pre-trained DNN. We validate the efficacy of SA-BMA in enhancing generalization performance in few-shot classification and distribution shift by ensuring flat posterior[1].

## 1 INTRODUCTION

Bayesian neural network (BNN) provides a theoretically grounded framework for modeling uncertainty in deep learning by approximating the posterior distribution of model parameters (MacKay, 1992b; Hinton & Van Camp, 1993; Neal, 2012). The approximated posterior is used for making predictions through Bayesian Model Averaging (BMA) (Wasserman, 2000; Fragoso et al., 2018; Wilson & Izmailov, 2020; Zeng & Van den Broeck, 2024). It allows BNNs to account for uncertainty in predictions, leading to more reliable outcomes compared to the deterministic neural network (DNN) (Kapoor et al., 2022; Kristiadi et al., 2022b). The accuracy and robustness of BNN predictions are heavily dependent on the quality of the approximated posterior (Kristiadi et al., 2022a; Wenzel et al., 2020).

One key factor influencing posterior quality is the flatness of the loss landscape. Flat modes of the loss landscape have been strongly associated with better generalization performance, as they represent solutions that are less sensitive to small perturbations in model parameters (Hochreiter & Schmidhuber, 1997; Keskar et al., 2016; Neyshabur et al., 2017). The concept of flatness has been extensively studied in the context of DNNs, but no comprehensive analysis has been conducted on its role in BNNs or its impact on BMA. The significance of flatness in BNNs has only been explored in a limited number of studies. SA-BNN (Nguyen et al., 2023) incorporated a flat-seeking optimizer into BNNs with a theoretical foundation. However, SA-BNN adapted a DNN-based optimizer to BNNs without considering the probabilistic nature of BNNs, resulting in only limited improvements. On the other hand, E-MCMC (Li & Zhang, 2023) introduced a guidance model to achieve flat posteriors, but this approach is less suited for large-scale models.

In this work, we first demonstrate that BNNs often struggle to capture the flatness. In detail, we compare the flatness of various BNN frameworks, including SWAG, VI, and MCMC, against that of DNNs. Furthermore, we show that BMA can be ineffective without ensuring flatness in the posterior distribution. These findings highlight the need for an optimization strategy that accounts for the probabilistic nature of BNNs to effectively estimate flat posteriors.

---

[1] Code for this paper is available in https://anonymous.4open.science/r/SA-BMA-A890.

Therefore, we propose Sharpness-Aware Bayesian Model Averaging (SA-BMA), a novel optimization approach that explicitly targets flat posterior distributions. We first compute the adversarial posterior that belongs to the vicinity of the current posterior through divergence, which maximizes the BNN loss function. After that, we update the posterior by employing the gradient of the adversarial posterior with respect to the BNN loss. We show that the proposed SA-BMA is an extended version of previous flatness-aware optimizers, Sharpness-aware Minimization (SAM) (Foret et al., 2020), Fisher SAM (FSAM) (Kim et al., 2022), and Natural Gradient (NG) (Amari, 1998) with specific conditions. Additionally, we propose a Bayesian Transfer Learning scheme integrated with SA-BMA, allowing for more efficient utilization of pre-trained models. We prove that SA-BMA improves the generalization performance of BNNs, particularly in few-shot classification and distribution shift scenarios, by ensuring flatness in the posterior.

Our major contributions are summarized as follows:

- We demonstrate that BNN often struggle to capture the flatness. Moreover, we show that BMA can be ineffective without flatness in the posterior.

- We suggest a Bayesian-fitting flat posterior seeking optimizer, SA-BMA. SA-BMA is a parameter space loss geometric optimizer, a generalized version of other loss geometric optimizers, such as SAM, FSAM, and NG.

- We propose a Bayesian Transfer Learning scheme integrated with SA-BMA to efficiently utilize pre-trained models. This scheme aims to enhance the generalization performance of BNN, especially in few-shot classification and distribution shifts, by ensuring posterior flatness.

## 2 PRELIMINARY

### 2.1 BAYESIAN NEURAL NETWORK

Bayesian neural network (BNN) aims to estimate the posterior distribution $p(w|\mathcal{D})$ of model parameters $w \subseteq \mathbb{R}^p$ with observed data points $\mathcal{D} = \{(x, y)\}$ with inputs $x$ and outputs $y$. The posterior distribution $p(w|\mathcal{D})$ is calculated by Bayes' Rule:

$$p(w|\mathcal{D}) = \frac{p(\mathcal{D}|w)p(w)}{\int_w p(\mathcal{D}|w)p(w)dw}, \tag{1}$$

where $p(\mathcal{D}|w)$ and $p(w)$ denote the likelihood of data $\mathcal{D}$ and the prior distribution over $w$, respectively. Due to the high dimensionality of neural networks, it is intractable to compute the marginal likelihood (evidence) of Eq. (1). Numerous studies have focused on approximating the posterior $p(w|\mathcal{D})$ with variational parameter $\theta \subseteq \mathbb{R}^q$ as $q_\theta(w|\mathcal{D})$, including Markov Chain Monte Carlo (MCMC) (Welling & Teh, 2011; Chen et al., 2014), Variational Inference (VI) (Graves, 2011; Ranganath et al., 2014; Blundell et al., 2015), and other variants employing DNN (MacKay, 1992a; Ritter et al., 2018; Daxberger et al., 2021a; Gal & Ghahramani, 2016; Maddox et al., 2019). Typically, model parameters are assumed to follow a Gaussian distribution $\mathcal{N}(\mu, \Sigma)$, where $\theta$ encompasses the mean $\mu$ and the covariance $\Sigma$.

Based on the approximated posterior, BNN makes predictions of the model on unobserved data $(x^*, y^*)$ through Bayesian Model Averaging (BMA):

$$p(y^*|x^*, \mathcal{D}) \approx \int_w p(y^*|x^*, w)q_\theta(w|\mathcal{D})dw \tag{2}$$

$$\approx \frac{1}{M}\sum_{m=1}^{M} p(y^*|x^*, w_m), \ \ w_m \sim q_\theta(w|\mathcal{D}), \tag{3}$$

where $M$ denotes the number of sampled models. Unfortunately, the integral in Eq. (2) is intractable. Monte Carlo integration (Eq. (3)) is a representative method to approximate posterior predictive. BNNs marginalize diverse solutions over the posterior of model parameters through BMA.

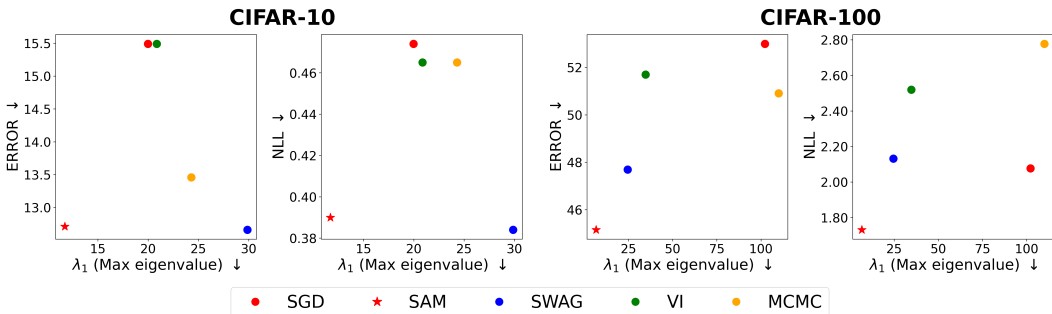

Figure 1: Flatness comparison between DNNs and BNNs. We measured the Error and NLL on CIFAR10 and CIFAR100. To assess flatness, we used the maximal Hessian eigenvalue of the loss, $\lambda_1$, where lower values indicate flatter models. BNN frameworks—VI, MCMC, and SWAG—were compared against DNNs trained with SGD and SAM, with SAM being a flatness-aware optimizer for DNNs. The results suggest that BNNs may not consistently capture flatness as effectively as DNNs.

## 2.2 FLATNESS AND OPTIMIZATION

Many studies have connected the flatness of loss surface and generalization (Hochreiter & Schmidhuber, 1994; 1997; Keskar et al., 2016; Neyshabur et al., 2017). The Hessian eigenvalue of loss is a widely adopted method for measuring the flatness of model, as smaller value indicates flatter regions in the loss landscape. However, due to the large size of neural networks, it is impractical to examine all eigenvalues. Therefore, maximal eigenvalue $\lambda_1(w)$ or ratio of eigenvalue $\lambda_1(w)/\lambda_5(w)$ is often used to compare flatness of model parameter, where $\lambda_i(w)$ denotes $i$-th maximal eigenvalue of model parameter $w$ (Keskar et al., 2016; Foret et al., 2020; Jastrzebski et al., 2020).

On the top of the connection between flatness and generalization, the local entropy (Baldassi et al., 2015; 2016) is one way to find flat minima. Typically, Entropy-SGD (Chaudhari et al., 2019) and Entropy-SGLD (Dziugaite & Roy, 2018) suggested finding flat modes by approximating the local entropy with a nested chain. On the other hand, SAM (Foret et al., 2020) aims to find parameters lie in the neighborhood $\gamma$ where the loss is consistently low by solving a min-max optimization problem. Within $\gamma$-ball neighborhood, the objective function of SAM is defined as:

$$l_{\text{SAM}}^{\gamma}(w) = \min_w \max_{\|\Delta w\|_p \leq \gamma} l(w + \Delta w),$$

where $l(\cdot)$ is the empirical loss function, such as cross-entropy loss in classification. $p$ is practically set to two, yielding $\Delta w = \gamma \nabla_w l(w)/\|\nabla_w l(w)\|_2$. On the one hand, FSAM (Kim et al., 2022) proposed replacing the Euclidean ball in SAM with a natural non-Euclidean ball induced by Fisher information:

$$l_{\text{FSAM}}^{\gamma}(w) = \min_w \max_{\|F_y(w)\Delta w\|_p \leq \gamma^2} l(w + \Delta w),$$

where the Fisher information matrix (FIM) is approximated as $F_y(w) = 1/|B|\nabla_w \log p(y|x, w)^2$ and $|B|$ denotes batch size. The Fisher inverse matrix is approximated as $F_y(w)^{-1} = 1/\sqrt{1 + \eta F_y(w)}$, with a hyperparameter $\eta$. This results in a closed-form perturbation, $\Delta w = \gamma \frac{(F_y(w)^{-1})^2 \nabla_w l(w)}{\|F_y(w)^{-1}\nabla_w l(w)\|_2}$, for $p = 2$. In other words, by preconditioning approximated FIM $F_y(w)$ over predictive distribution, FSAM attempts to find curvature-aware perturbation. SAM and FSAM are derived under deterministic $w$, and the $F_y(w)$ is defined in predictive distribution $p(y|x, w)$, not in parameter space.

## 3 FLATNESS DOES MATTER FOR BAYESIAN MODEL AVERAGING

We first cast a question of whether BNNs inherently captures the flatness. To answer this, we compare the flatness of DNNs and BNNs without flatness-aware optimizer and demonstrate empirically that BNNs often struggle to capture the flatness. We also show flatness does matter for BMA both experimentally and theoretically.

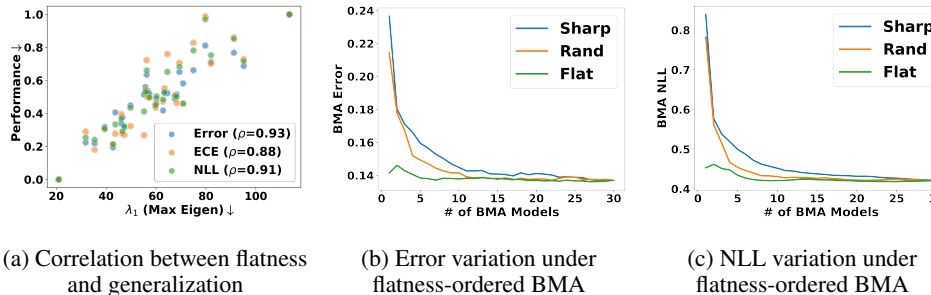

(a) Correlation between flatness and generalization

(b) Error variation under flatness-ordered BMA

(c) NLL variation under flatness-ordered BMA

Figure 2: (a) illustrates the clear correlation between flatness, the maximal Hessian eigenvalue ($\lambda_1$), and normalized generalization metrics such as classification error, ECE, and NLL. $\rho$ represents the Pearson correlation coefficient. We conjecture that the flatness is crucial for the generalization performance of BNN. (b)-(c) represent the variation of generalization performance under flatness-ordered BMA. "Flat" denotes starting model averaging from the flattest model, and "Sharp" means the opposite of "Flat". "Rand" denotes starting BMA from a random sample of prepared 30 models. This result shows degradation or stagnation can appear without considering the flatness in BMA and highlights the need to account for flatness in BNNs.

### 3.1 BNN STRUGGLES TO CAPTURE THE FLATNESS

We first investigate whether BNNs effectively capture flatness and find that they often struggle to do so. To assess flatness, we use $\lambda_1(w)$ and $\lambda_1(w)/\lambda_5(w)$, as criteria (Foret et al., 2020; Li & Zhang, 2023). For BMA, we compute the average of these metrics across $M$ model samples $w_m, m = 1, ..., M$, as $1/M \sum_{m=1}^{M} \lambda_1(w_m)$ and $1/M \sum_{m=1}^{M} \lambda_1(w_m)/\lambda_5(w_m)$ (detailed in Appendix A.1). We simply refer to these criteria as $\lambda_1$ and $\lambda_1/\lambda_5$. We also measure the Error ($100 - $ Accuracy), Expected Calibration Error (ECE) (Guo et al., 2017), and Negative Log Likelihood (NLL) to compare the generalization ability. We mainly set ResNet18 (RN18) (He et al., 2016) without Batch Normalization (BN) (Ioffe & Szegedy, 2015) as a backbone model. For BNN frameworks, we adopt VI, MCMC, and SWAG. To minimize the effect on measuring the flatness, we remove the BN and also do not adjust data augmentation. We present more detailed results of the analysis between flatness and performance across various settings in Appendix A.2, demonstrating consistent outcomes.

In Figure 1, BNNs do not consistently capture the loss geometry well. BNNs trained with SGD often exhibit higher $\lambda_1$ compared to DNN trained with SGD. They also show significantly higher $\lambda_1$ when compared to DNN trained with SAM. This is consistent with Figure 7 and 8 (Appendix A.2), trained on diverse learning rate schedulers. Therefore, we conclude that BNNs often struggle to capture flatness, raising the question of whether BMA, with its unique prediction approach, is also affected by flatness.

### 3.2 BMA CAN BE INEFFECTIVE WITHOUT FLATNESS

BNNs possess the benefit of BMA, which anticipates performance enhancement through model ensemble based on the approximated posterior distribution. However, following the fact that BNNs often struggle to capture the flatness, we cast another question: "Does the flatness also affect BMA?"

First, we consolidate that the correlation between flatness and generalization performance also exists in BMA. Figure 2a shows the affirmative correlation between performance and flatness throughout 30 sampled model parameters from posterior trained on CIFAR10 with RN18 w/o BN. We provide additional plots in Figure 9 (Appendix A.3), demonstrating consistent results.

Second, we observe how performance changes when the sampling model parameter in BMA is performed based on flatness. We first sample 30 models trained on CIFAR10 with RN18 w/o BN. Then, we start to do BMA through three criteria. "Flat" denotes starting model averaging from the flattest model, and "Sharp" means the opposite of "Flat". "Rand" denotes starting BMA from a random sample of prepared 30 models. From Figure 2b and 2c, we conclude that degradation or stagnation can appear without considering the flatness during BMA. Figure 10 and 11 (Appendix A.4) support the conclusion, as well. With a closer look at the "Flat" label, where progressively sharper

models are contained, we observe that performances do not improve as the number of averaged models increases. These points prove the necessity of flatness in BMA.

Along with the experimental influence of flatness on BMA, we also show that the individual sampled models affect the flatness of BMA. Specifically, we suggest a theoretical bound of flatness on BMA through the Hessian eigenvalue of loss. The Hessian eigenvalue of loss is a typical measurement to compare the flatness of neural networks. BMA marginalizes diverse predictions by ensembling model output. The loss of the weight-averaged model is approximately the same as the loss of the ensemble model, with a small error term based on the difference between their outputs (Lemma 1 in Appendix E.1) (Izmailov et al., 2018; Wortsman et al., 2022; Rame et al., 2022). We demonstrate the flatness bound for WA and connect it to that of BMA. Through Weyl's inequality (Weyl, 1912), the bound of the Hessian eigenvalue of $w_{\text{WA}} = 1/M \sum_{m=1}^{M} w_m$ is defined as:

**Theorem 1.** *With $M$ model sample $w_m, m = 1, ..., M$, the maximal eigenvalue of averaged Hessian of loss $\lambda_{max}(H_{w_{\text{WA}}})$ is bounded as follow:*

$$\max \left( \left\{ \frac{1}{M} \left( \lambda_{\max}(H_{w_m}) + \sum_{\substack{n=1 \\ n \neq m}}^{M} \lambda_{\min}(H_{w_n}) \right) \right\}_{m=1}^{M} \right) \leq \lambda_{\max}(H_{w_{\text{WA}}}) \leq \frac{\sum_{m=1}^{M} \lambda_{\max}(H_{w_m})}{M}.$$

Theorem 1 implies that the flatness of BMA reflects the flatness of model samples. If a sampled model had a large Hessian eigenvalue, the lower bound of Hessian eigenvalue can be larger. Namely, the ensembled model can be located in a sharp region by ensembling sharp model samples from posterior. Through empirical and theoretical analysis of flatness in BNNs, we confirm that a flat posterior is necessary to ensure the individual sampled models are flat, leading to more effective BMA.

# 4 BAYESIAN MODEL AVERAGING WITH FLAT POSTERIOR

For more effective BMA, we propose a Bayesian flat-seeking optimizer (Section 4.1) and Bayesian transfer learning combined with diverse BNN frameworks (Section 4.2).

## 4.1 BAYESIAN FLAT-SEEKING OPTIMIZER

To deal with the probabilistic nature of BNN, we suggest a new objective function based on VI:

$$l_{\text{SA-BMA}}^{\gamma}(\theta) = \min_{\theta} \max_{d|\theta+\Delta\theta,\theta| \leq \gamma^2} l(\theta + \Delta\theta) + \beta D_{\text{KL}}[q_\theta(w|\mathcal{D})||p(w)] \tag{4}$$

$$\text{s.t. } d|\theta + \Delta\theta, \theta| = D_{\text{KL}}\big[q_{\theta+\Delta\theta}(w|\mathcal{D}) \,||\, q_\theta(w|\mathcal{D})\big], \tag{5}$$

where $\theta$ and $\Delta\theta$ denote the variational parameters and perturbation on them, respectively. $l(\cdot)$ denotes empirical loss, such as NLL under $q_\theta(w|\mathcal{D})$, and $\beta$ is a hyperparameter that controls the influence of the prior.

Through the relationship between KL divergence and FIM, the objective function (Eq. (4)) is rewritten as:

$$l_{\text{SA-BMA}}^{\gamma}(\theta) = \min_{\theta} \max_{\Delta\theta^T F_\theta(\theta)\Delta\theta \leq \gamma^2} l(\theta + \Delta\theta) + \beta D_{\text{KL}}[q_\theta(w|\mathcal{D})||p(w)], \tag{6}$$

where $F_\theta(\theta) = \mathbb{E}_{w,\mathcal{D}}[\nabla_\theta \log q_\theta(w|\mathcal{D}) \nabla_\theta \log q_\theta(w|\mathcal{D})^T]$. Note the FIM is defined over parameters. Accordingly, the adversarial posterior is directly obtained in the parameter space, enabling BNNs to better construct neighbor to flat minima.

In first step, we get the closed-form $\Delta\theta_{\text{SA-BMA}}$ ($\Delta\theta$ in Eq. (6)) as:

$$\Delta\theta_{\text{SA-BMA}} = \gamma \frac{F_\theta(\theta)^{-1} \nabla_\theta l(\theta)}{\sqrt{\nabla_\theta l(\theta)^T F_\theta(\theta)^{-1} \nabla_\theta l(\theta)}}. \tag{7}$$

Detailed formula derivation of the optimal perturbation for SA-BMA (Eq. (7)) is provided in Appendix E.2.

In second step, we update $\theta$ by gradients from adversarial posterior perturbated with $\Delta\theta_{\text{SA-BMA}}$:

$$\nabla_\theta l^\gamma_{\text{SA-BMA}}(\theta) \approx \nabla_\theta l(\theta + \Delta\theta_{\text{SA-BMA}}). \tag{8}$$

SA-BMA enhances the estimation of flat posteriors by adapting SAM to the probabilistic nature of BNNs, defining adversarial perturbation ball directly in the parameter space.

**Generalized version of geometric optimizers**   Notably, SA-BMA is a generalized version of SAM, FSAM, and NG under deterministic parameters, as shown in Theorem 2. Proof of Theorem 2 are provided in Appendix E.3.

**Theorem 2.** *(Informal) Suppose the model parameter $w$ is deterministic and the loss function $l(\cdot)$ is twice continuously differentiable. Let $\gamma' = \gamma/\sqrt{\nabla_\theta l(\theta)^T F_\theta(\theta)^{-1} \nabla_\theta l(\theta)}$, then*

>  *i)  SA-BMA degenerates to FSAM by using the diagonal terms of FIM.*
>
>  *ii)  SA-BMA degenerates to SAM if FIM is an identity matrix.*
>
>  *iii)  Update rule of SA-BMA $\theta - \eta_{SA\text{-}BMA}\nabla_\theta l(\theta + \Delta\theta)$ degenerates to update rule of NG $\theta - \eta_{NG}F_y(\theta)^{-1}\nabla_\theta l(\theta)$ with learning rate $\eta_{SA\text{-}BMA} = \frac{\eta_{NG}}{(1+\gamma')}F_\theta(\theta)^{-1}$.*

SA-BMA replaces the output space FIM, used by existing DNN-based flat-seeking optimizers, with the parameter space FIM. By taking into account the off-diagonal terms of the FIM, SA-BMA more accurately estimates flatness, leading to more precise optimization. This makes SA-BMA better suited for BNNs. Additionally, SA-BMA becomes equivalent to NG when using a specific learning rate, suggesting the potential for accelerating convergence.

## 4.2 Bayesian Transfer Learning

The proposed optimizer theoretically captures curvature more exactly, but it faces practical limitations when applied to neural networks with a large number of parameters. To address this limitation, we propose a Bayesian transfer learning scheme alongside the proposed optimizer. In this scheme, we leverage pre-trained DNN as a prior and train only a subset of the model's parameters, enhancing scalability while maintaining performance. The proposed Bayesian transfer learning scheme consists of three steps, and Algorithm 1 (Appendix D.3) depicts how it operates.

First, we load a pre-trained DNN $w_{\text{MAP}} \subseteq \mathbb{R}^p$ optimized by Maximum A Posteriori (MAP). We change the loaded DNN into BNN on the source or downstream task $\mathcal{D}^{\text{pr}}$. Formally, we get $q^{\text{pr}}_\theta(w|\mathcal{D}^{\text{pr}})$. Various BNN frameworks, such as VI, SWAG, and others, can be employed to transform a DNN into a BNN. This study mainly employs MCMC, SWAG (Maddox et al., 2019), and MOPED (Krishnan et al., 2020) for VI. Note that the proposed optimizer is not employed in this step.

Second, we set the converted BNN as prior and initial points and train new posterior with the proposed optimizer, following objective function:

$$l^\gamma_{\text{SA-BMA}}(\theta) = \min_\theta \max_{\Delta\theta^T F_\theta(\theta)\Delta\theta \leq \gamma^2} l(\theta + \Delta\theta) + \beta D_{\text{KL}}[q_\theta(w|\mathcal{D}^{\text{ft}})||q^{\text{pr}}_\theta(w|\mathcal{D}^{\text{pr}})], \tag{9}$$

where $\mathcal{D}^{\text{ft}}$ represents the downstream dataset used for fine-tuning, and $l(\cdot)$ denotes the empirical loss function. To reduce computational complexity, we also incorporate a subnetwork BNN strategy, which has been extensively explored in recent studies (Izmailov et al., 2020; Daxberger et al., 2021c; Sharma et al., 2023; Snoek et al., 2015; Daxberger et al., 2021b; Harrison et al., 2023). In this work, we set the trainable parameters to parameters of normalization and last layers. Additionally, we reinitialize the last layer using a simple Gaussian distribution $\mathcal{N}(0, \alpha I)$ during fine-tuning on the downstream dataset, where $\alpha$ is a hyperparameter controlling the variance. This approach is expected to facilitate scalable and stable training by leveraging pre-trained DNNs.

## 5 Experiments

We first verify that SA-BMA effectively converges to flat minima using a synthetic example (Section 5.1). Then, we demonstrate that training from scratch with SA-BMA optimizer leads to improved generalization performance (Section 5.2). In addition, by integrating Bayesian transfer learning, we

show that SA-BMA performs well in few-shot image classification tasks (Section 5.3) and is robust to distribution shifts (Section 5.4). Finally, we visualize the loss surface and compare the Hessian eigenvalues numerically, further confirming that the model approximates flat posteriors (Section 5.5).

## 5.1 SYNTHETIC EXAMPLE

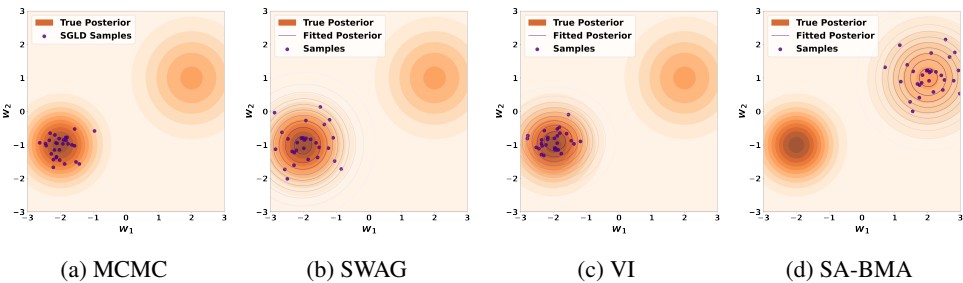

|        (a) MCMC        |        (b) SWAG        |         (c) VI         |       (d) SA-BMA       |

Figure 3: Posterior approximation with synthetic example. When both flat and sharp modes coexist, we compared how optimizers approximate the posterior. Unlike other methods, the proposed SA-BMA converged to the flat mode, demonstrating its effectiveness in finding more stable solutions.

We conduct a toy experiment designed to create sharp and flat modes. We run various methods, including SGD, MCMC with SGLD, SWAG, and VI, to compare them with SA-BMA combined with VI and SWAG. The results in Figure 3 and 12 (Appendix B) show that the baseline methods based on SGD and SGLD converge to the sharp mode without any consideration for flatness. In contrast, the proposed SA-BMA consistently converged to the flat mode, regardless of the BNN framework combined. We provide the setting and additional results in Appendix B.

## 5.2 LEARNING FROM SCRATCH

Table 1: Performance of learning from scratch with RN18 and modified ViT-B/16†. SA-BMA (VI), SA-BMA (MCMC), and SA-BMA (SWAG) indicate the specific BNN framework combined with SA-BMA. **Bold** highlights the best performance within each BNN framework, while red indicates the overall best performance across all frameworks. SA-BMA leads better performance across all BNN frameworks and shows superior performance both on the CIFAR10 and CIFAR100.

| Backbone | RN18 | | | | | | ViT-B/16† | | | | | |
|---|---|---|---|---|---|---|---|---|---|---|---|---|
| Dataset | CIFAR10 | | | CIFAR100 | | | CIFAR10 | | | CIFAR100 | | |
| Method | ACC ↑ | ECE ↓ | NLL ↓ | ACC ↑ | ECE ↓ | NLL ↓ | ACC ↑ | ECE ↓ | NLL ↓ | ACC ↑ | ECE ↓ | NLL ↓ |
| SGD | $83.28_{\pm0.49}$ | $0.058_{\pm0.005}$ | $0.540_{\pm0.006}$ | $50.33_{\pm0.62}$ | $0.123_{\pm0.016}$ | $1.976_{\pm0.055}$ | $81.20_{\pm1.31}$ | $0.050._{\pm0.002}$ | $0.569_{\pm0.027}$ | $48.66_{\pm0.21}$ | $0.062_{\pm0.013}$ | $1.956_{\pm0.021}$ |
| SAM | $\mathbf{87.59}_{\pm3.10}$ | $\mathbf{0.031}_{\pm0.017}$ | $\mathbf{0.389}_{\pm0.065}$ | $51.48_{\pm0.05}$ | $0.096_{\pm0.026}$ | $1.873_{\pm0.042}$ | $81.25_{\pm0.10}$ | $\mathbf{0.020}_{\pm0.003}$ | $\mathbf{0.550}_{\pm0.002}$ | $54.91_{\pm4.20}$ | $0.053_{\pm0.020}$ | $1.709_{\pm0.148}$ |
| FSAM | $83.38_{\pm0.86}$ | $0.052_{\pm0.003}$ | $0.540_{\pm0.010}$ | $50.87_{\pm1.29}$ | $0.114_{\pm0.008}$ | $1.963_{\pm0.058}$ | $\mathbf{81.57}_{\pm1.49}$ | $0.046_{\pm0.006}$ | $0.563_{\pm0.036}$ | $48.75_{\pm0.42}$ | $0.055_{\pm0.010}$ | $1.956_{\pm0.003}$ |
| bSAM | $84.28_{\pm0.32}$ | $0.051_{\pm0.010}$ | $0.502_{\pm0.012}$ | $\mathbf{52.55}_{\pm0.30}$ | $\mathbf{0.087}_{\pm0.011}$ | $\mathbf{1.802}_{\pm0.027}$ | $80.33_{\pm0.88}$ | $0.037_{\pm0.007}$ | $0.588_{\pm0.012}$ | $\mathbf{57.75}_{\pm0.29}$ | $\mathbf{0.040}_{\pm0.014}$ | $\mathbf{1.573}_{\pm0.015}$ |
| VI | $82.61_{\pm0.51}$ | $0.067_{\pm0.003}$ | $0.632_{\pm0.008}$ | $51.45_{\pm0.32}$ | $0.037_{\pm0.007}$ | $1.874_{\pm0.007}$ | $75.81_{\pm0.88}$ | $0.027_{\pm0.021}$ | $0.715_{\pm0.038}$ | $48.97_{\pm0.20}$ | $0.037_{\pm0.012}$ | $1.965_{\pm0.004}$ |
| **SA-BMA (VI)** | $\mathbf{85.34}_{\pm0.18}$ | $\mathbf{0.028}_{\pm0.006}$ | $\mathbf{0.431}_{\pm0.001}$ | $\mathbf{54.49}_{\pm0.82}$ | $\color{red}\mathbf{0.016}_{\pm0.003}$ | $\mathbf{1.699}_{\pm0.021}$ | $\mathbf{76.23}_{\pm0.44}$ | $\mathbf{0.018}_{\pm0.006}$ | $\mathbf{0.692}_{\pm0.010}$ | $\mathbf{51.62}_{\pm1.12}$ | $0.038_{\pm0.013}$ | $\mathbf{1.884}_{\pm0.026}$ |
| MCMC | $84.82_{\pm0.13}$ | $0.049_{\pm0.001}$ | $0.523_{\pm0.008}$ | $58.38_{\pm0.16}$ | $0.090_{\pm0.002}$ | $1.742_{\pm0.014}$ | $81.80_{\pm0.46}$ | $0.014_{\pm0.003}$ | $0.542_{\pm0.023}$ | $51.79_{\pm0.29}$ | $0.081_{\pm0.001}$ | $2.068_{\pm0.016}$ |
| E-MCMC | $85.45_{\pm0.27}$ | $0.037_{\pm0.002}$ | $0.479_{\pm0.006}$ | $60.38_{\pm0.21}$ | $0.074_{\pm0.003}$ | $1.574_{\pm0.002}$ | $81.97_{\pm0.49}$ | $0.034_{\pm0.004}$ | $0.545_{\pm0.014}$ | $50.48_{\pm0.13}$ | $0.068_{\pm0.005}$ | $2.010_{\pm0.007}$ |
| **SA-BMA (MCMC)** | $\mathbf{86.98}_{\pm0.19}$ | $\mathbf{0.030}_{\pm0.004}$ | $\mathbf{0.393}_{\pm0.001}$ | $\mathbf{61.94}_{\pm0.37}$ | $\mathbf{0.029}_{\pm0.003}$ | $\mathbf{1.467}_{\pm0.006}$ | $\mathbf{82.49}_{\pm1.95}$ | $\color{red}\mathbf{0.012}_{\pm0.003}$ | $\mathbf{0.528}_{\pm0.067}$ | $\color{red}\mathbf{61.10}_{\pm1.44}$ | $\mathbf{0.046}_{\pm0.005}$ | $\color{red}\mathbf{1.461}_{\pm0.067}$ |
| SWAG | $88.95_{\pm0.09}$ | $0.044_{\pm0.015}$ | $0.349_{\pm0.013}$ | $59.48_{\pm0.19}$ | $\mathbf{0.030}_{\pm0.002}$ | $1.594_{\pm0.011}$ | $83.70_{\pm0.30}$ | $0.044_{\pm0.011}$ | $0.493_{\pm0.020}$ | $54.76_{\pm2.20}$ | $0.151_{\pm0.025}$ | $2.008_{\pm0.136}$ |
| F-SWAG | $89.35_{\pm0.19}$ | $0.028_{\pm0.013}$ | $0.323_{\pm0.010}$ | $60.44_{\pm0.20}$ | $0.074_{\pm0.023}$ | $1.566_{\pm0.006}$ | $83.57_{\pm0.41}$ | $0.046_{\pm0.015}$ | $0.498_{\pm0.029}$ | $56.80_{\pm1.44}$ | $0.061_{\pm0.017}$ | $1.733_{\pm0.073}$ |
| **SA-BMA (SWAG)** | $\color{red}\mathbf{89.84}_{\pm0.30}$ | $\color{red}\mathbf{0.019}_{\pm0.002}$ | $\color{red}\mathbf{0.306}_{\pm0.006}$ | $\color{red}\mathbf{63.63}_{\pm0.60}$ | $0.052_{\pm0.007}$ | $\color{red}\mathbf{1.342}_{\pm0.003}$ | $\color{red}\mathbf{84.44}_{\pm0.58}$ | $\mathbf{0.028}_{\pm0.008}$ | $\color{red}\mathbf{0.464}_{\pm0.011}$ | $\mathbf{57.64}_{\pm1.42}$ | $\color{red}\mathbf{0.032}_{\pm0.005}$ | $1.590_{\pm0.050}$ |

We evaluate the performance of SA-BMA by combining it with various BNN frameworks, VI, MCMC, and SWAG. For this, we train both RN18 and ViT-B/16† models from scratch in CIFAR10 and CIFAR100 (Krizhevsky et al., 2009). We use modified ViT-B/16† (Dosovitskiy et al., 2020) to deal with the underfitting issue over small dataset of ViT-B/16 (Liu et al., 2021; Zhu et al., 2023a). SA-BMA was applied only to the normalization and last layers, while all other layers were trained using SGD. We compare the performance of SA-BMA with DNNs trained using SGD, SAM, and FSAM, as well as BNN frameworks—SWAG, VI, and MCMC—and prior methods like F-SWAG (Nguyen et al., 2023), bSAM (Möllenhoff & Khan, 2022), and E-MCMC (Li & Zhang, 2023). For MCMC and E-MCMC, we consistently use SGLD in all experiments, evaluating models based on accuracy (ACC), ECE, and NLL. As shown in Table 1, SA-BMA consistently improved performance when integrated

with all BNN frameworks, demonstrating superior results compared to all baselines. Experimental details are provided in Appendix C.

## 5.3 FEW-SHOT IMAGE CLASSIFICATION WITH BAYESIAN TRANSFER LEARNING

We also evaluate the performance of the proposed SA-BMA on a few-shot image classification task in the context of transfer learning. In this experiment, we adopt RN18 and ViT-B/16 pre-trained on ImageNet (IN) 1K (Russakovsky et al., 2015) as backbone. We add baselines for the Bayesian transfer learning baseline MOPED (Krishnan et al., 2020) and Pre-Train Your Loss (PTL) (Shwartz-Ziv et al., 2022) in this setting. Detailed configuration for experiments is provided in Appendix D.

First, we evaluate our model in CIFAR10 and CIFAR100 (Krizhevsky et al., 2009) with ten images per class, each. As illustrated in Table 2, SA-BMA with diverse BNN frameworks consistently outperforms existing baselines in terms of both accuracy and uncertainty quantification. Unlike scratch learning, SA-BMA (VI) outperforms SA-BMA (SWAG) in few-shot image classification tasks. This can be attributed to the nature of few-shot tasks, where VI, which only learns a diagonal covariance, is less prone to underfitting due to the limited amount of data.

Table 2: Downstream task performance with RN18 and ViT-B/16 pre-trained on IN 1K. **Bold** highlights the best performance within each BNN framework, while red indicates the overall best performance across all frameworks. SA-BMA shows superior performance both on the CIFAR10 and CIFAR100 10-shot, with the sole exception being the ECE on the CIFAR100 10-shot in RN18.

| Backbone | RN18 | | | | | | ViT-B/16 | | | | | |
|---|---|---|---|---|---|---|---|---|---|---|---|---|
| Dataset | C10 10-shot | | | C100 10-shot | | | C10 10-shot | | | C100 10-shot | | |
| Method | ACC ↑ | ECE ↓ | NLL ↓ | ACC ↑ | ECE ↓ | NLL ↓ | ACC ↑ | ECE ↓ | NLL ↓ | ACC ↑ | ECE ↓ | NLL ↓ |
| SGD | $55.52_{\pm0.32}$ | $\mathbf{0.062}_{\pm0.006}$ | $1.302_{\pm0.020}$ | $44.29_{\pm0.83}$ | $\mathbf{0.025}_{\pm0.005}$ | $2.133_{\pm0.043}$ | $84.37_{\pm1.47}$ | $0.056_{\pm0.061}$ | $0.503_{\pm0.038}$ | $68.78_{\pm0.21}$ | $0.143_{\pm0.007}$ | $\mathbf{1.193}_{\pm0.019}$ |
| SAM | $56.54_{\pm2.57}$ | $0.129_{\pm0.013}$ | $1.354_{\pm0.089}$ | $\mathbf{44.51}_{\pm0.07}$ | $0.065_{\pm0.007}$ | $\mathbf{2.089}_{\pm0.013}$ | $84.35_{\pm0.81}$ | $\mathbf{0.035}_{\pm0.012}$ | $\mathbf{0.486}_{\pm0.023}$ | $\mathbf{68.93}_{\pm0.37}$ | $0.153_{\pm0.005}$ | $1.200_{\pm0.021}$ |
| FSAM | $54.04_{\pm4.11}$ | $0.139_{\pm0.010}$ | $1.432_{\pm0.068}$ | $44.07_{\pm1.21}$ | $0.056_{\pm0.006}$ | $2.159_{\pm0.064}$ | $\mathbf{84.51}_{\pm0.50}$ | $0.073_{\pm0.085}$ | $0.517_{\pm0.061}$ | $68.74_{\pm0.39}$ | $\mathbf{0.110}_{\pm0.007}$ | $1.166_{\pm0.024}$ |
| bSAM | $\mathbf{56.56}_{\pm1.18}$ | $0.083_{\pm0.006}$ | $\mathbf{1.280}_{\pm0.027}$ | $43.93_{\pm0.48}$ | $0.060_{\pm0.003}$ | $2.167_{\pm0.026}$ | $82.85_{\pm2.10}$ | $0.113_{\pm0.008}$ | $0.583_{\pm0.062}$ | $68.42_{\pm0.40}$ | $0.148_{\pm0.019}$ | $1.219_{\pm0.031}$ |
| MOPED | $57.29_{\pm1.20}$ | $0.093_{\pm0.006}$ | $1.297_{\pm0.045}$ | $44.30_{\pm0.42}$ | $0.047_{\pm0.006}$ | $2.127_{\pm0.005}$ | $84.50_{\pm1.36}$ | $\mathbf{0.023}_{\pm0.009}$ | $0.474_{\pm0.038}$ | $68.80_{\pm0.77}$ | $0.111_{\pm0.001}$ | $1.165_{\pm0.029}$ |
| SA-BMA (VI) | $\mathbf{64.98}_{\pm1.37}$ | $\mathbf{0.016}_{\pm0.007}$ | $\mathbf{0.997}_{\pm0.046}$ | $\mathbf{49.09}_{\pm1.38}$ | $0.071_{\pm0.004}$ | $\mathbf{1.893}_{\pm0.036}$ | $\mathbf{87.56}_{\pm1.10}$ | $0.044_{\pm0.012}$ | $\mathbf{0.397}_{\pm0.026}$ | $\mathbf{71.37}_{\pm0.36}$ | $\mathbf{0.060}_{\pm0.007}$ | $\mathbf{1.023}_{\pm0.012}$ |
| MCMC | $56.31_{\pm1.27}$ | $0.083_{\pm0.003}$ | $1.305_{\pm0.063}$ | $44.28_{\pm0.95}$ | $0.021_{\pm0.002}$ | $2.155_{\pm0.038}$ | $83.93_{\pm1.33}$ | $0.069_{\pm0.010}$ | $0.523_{\pm0.039}$ | $66.48_{\pm1.18}$ | $0.077_{\pm0.011}$ | $1.224_{\pm0.044}$ |
| PTL | $57.26_{\pm1.44}$ | $0.116_{\pm0.003}$ | $1.345_{\pm0.004}$ | $43.00_{\pm1.05}$ | $0.120_{\pm0.006}$ | $2.383_{\pm0.062}$ | $\mathbf{85.76}_{\pm1.37}$ | $0.080_{\pm0.014}$ | $0.482_{\pm0.027}$ | $65.52_{\pm2.45}$ | $\mathbf{0.056}_{\pm0.006}$ | $1.260_{\pm0.095}$ |
| E-MCMC | $56.69_{\pm2.14}$ | $0.142_{\pm0.004}$ | $1.266_{\pm0.054}$ | $41.57_{\pm0.04}$ | $0.046_{\pm0.012}$ | $2.370_{\pm0.175}$ | $83.91_{\pm1.16}$ | $0.333_{\pm0.010}$ | $0.877_{\pm0.044}$ | $63.40_{\pm0.01}$ | $0.280_{\pm0.008}$ | $1.655_{\pm0.024}$ |
| SA-BMA (MCMC) | $\mathbf{57.49}_{\pm0.64}$ | $\mathbf{0.039}_{\pm0.00}$ | $\mathbf{1.248}_{\pm0.045}$ | $\mathbf{45.72}_{\pm0.56}$ | $\mathbf{0.016}_{\pm0.003}$ | $\mathbf{2.062}_{\pm0.050}$ | $84.82_{\pm1.84}$ | $\mathbf{0.051}_{\pm0.018}$ | $\mathbf{0.449}_{\pm0.048}$ | $\mathbf{68.73}_{\pm1.09}$ | $0.061_{\pm0.004}$ | $\mathbf{1.117}_{\pm0.042}$ |
| SWAG | $56.31_{\pm0.60}$ | $0.094_{\pm0.004}$ | $1.315_{\pm0.056}$ | $44.14_{\pm1.28}$ | $\mathbf{0.034}_{\pm0.010}$ | $2.161_{\pm0.058}$ | $83.51_{\pm2.22}$ | $0.022_{\pm0.015}$ | $0.510_{\pm0.072}$ | $68.72_{\pm0.45}$ | $0.065_{\pm0.005}$ | $1.136_{\pm0.014}$ |
| F-SWAG | $57.65_{\pm1.20}$ | $0.075_{\pm0.003}$ | $1.249_{\pm0.038}$ | $46.09_{\pm0.44}$ | $0.062_{\pm0.006}$ | $2.089_{\pm0.002}$ | $83.87_{\pm1.28}$ | $0.013_{\pm0.005}$ | $0.492_{\pm0.040}$ | $68.84_{\pm0.77}$ | $0.076_{\pm0.012}$ | $1.137_{\pm0.020}$ |
| SA-BMA (SWAG) | $\mathbf{61.79}_{\pm4.34}$ | $\mathbf{0.026}_{\pm0.004}$ | $\mathbf{1.214}_{\pm0.119}$ | $\mathbf{47.45}_{\pm0.60}$ | $0.055_{\pm0.018}$ | $\mathbf{2.044}_{\pm0.022}$ | $\mathbf{86.81}_{\pm0.78}$ | $\mathbf{0.010}_{\pm0.003}$ | $\mathbf{0.399}_{\pm0.034}$ | $\mathbf{70.10}_{\pm0.18}$ | $\mathbf{0.045}_{\pm0.015}$ | $\mathbf{1.063}_{\pm0.023}$ |

Table 3: Downstream task accuracy with RN50 and ViT-B/16 pre-trained on IN 1K. SA-BMA (SWAG) denotes using SWAG to convert pre-trained model into BNN. **Bold** and underline denote best and second best performance each. SA-BMA demonstrates superior performance across all 16-shot datasets.

| Backbone | RN50 | | | | | ViT-B/16 | | | | |
|---|---|---|---|---|---|---|---|---|---|---|
| Method | EuroSAT | Flowers102 | Pets | UCF101 | Avg | EuroSAT | Flowers102 | Pets | UCF101 | Avg |
| SGD | $86.75_{\pm1.47}$ | $93.16_{\pm0.27}$ | $89.95_{\pm0.51}$ | $66.34_{\pm0.59}$ | $84.05_{\pm0.33}$ | $81.25_{\pm1.03}$ | $91.24_{\pm0.03}$ | $88.68_{\pm0.92}$ | $68.64_{\pm0.51}$ | $82.45_{\pm0.56}$ |
| SAM | $87.85_{\pm0.49}$ | $94.80_{\pm0.17}$ | $90.23_{\pm0.78}$ | $70.40_{\pm0.76}$ | $85.82_{\pm0.25}$ | $82.53_{\pm0.65}$ | $\underline{93.08}_{\pm0.87}$ | $\underline{90.66}_{\pm0.74}$ | $\underline{70.66}_{\pm1.03}$ | $\underline{84.23}_{\pm0.60}$ |
| SWAG | $88.97_{\pm1.56}$ | $93.27_{\pm0.15}$ | $89.95_{\pm0.46}$ | $66.41_{\pm0.30}$ | $84.65_{\pm0.37}$ | $81.62_{\pm0.66}$ | $91.21_{\pm0.91}$ | $88.67_{\pm0.42}$ | $67.65_{\pm0.45}$ | $82.29_{\pm0.31}$ |
| F-SWAG | $90.03_{\pm1.08}$ | $\underline{94.84}_{\pm0.26}$ | $\underline{90.12}_{\pm0.57}$ | $\underline{70.00}_{\pm0.87}$ | $\underline{86.25}_{\pm0.19}$ | $82.72_{\pm0.49}$ | $92.93_{\pm0.93}$ | $90.60_{\pm0.55}$ | $68.67_{\pm0.39}$ | $83.73_{\pm0.35}$ |
| MOPED | $85.21_{\pm3.14}$ | $92.15_{\pm0.73}$ | $89.25_{\pm0.61}$ | $65.85_{\pm0.99}$ | $83.11_{\pm0.86}$ | $\underline{83.97}_{\pm0.49}$ | $91.71_{\pm0.87}$ | $89.90_{\pm0.54}$ | $69.66_{\pm0.53}$ | $83.81_{\pm0.51}$ |
| PTL | $\underline{90.01}_{\pm0.39}$ | $92.55_{\pm0.53}$ | $89.43_{\pm0.41}$ | $65.00_{\pm1.24}$ | $84.25_{\pm0.30}$ | $83.76_{\pm0.61}$ | $88.43_{\pm1.27}$ | $88.54_{\pm0.53}$ | $60.38_{\pm1.84}$ | $80.28_{\pm0.03}$ |
| SA-BMA | $\mathbf{90.16}_{\pm1.04}$ | $\mathbf{95.85}_{\pm1.26}$ | $\mathbf{90.23}_{\pm0.58}$ | $\mathbf{71.57}_{\pm0.27}$ | $\mathbf{86.95}_{\pm0.65}$ | $\mathbf{84.60}_{\pm0.25}$ | $\mathbf{94.15}_{\pm0.80}$ | $\mathbf{91.30}_{\pm0.25}$ | $\mathbf{72.63}_{\pm1.12}$ | $\mathbf{85.67}_{\pm0.14}$ |

We also conduct extra experiments on four fine-grained image classification benchmarks, EuroSAT (Helber et al., 2019), Flowers102 (Nilsback & Zisserman, 2008), Pets (Parkhi et al., 2012), and UCF101 (Soomro et al., 2012). From this point, we conduct all experiments using SA-BMA with SWAG for efficiency. We observe that SA-BMA achieves the best accuracy (Table 3) and NLL (Table 15 in Appendix F) across all datasets, as well. We demonstrate that SA-BMA syner-

Table 4: Downstream task accuracy of CLIP with visual encoder, RN50 and ViT-B/16. **Bold** and underline denote best and second best performance each. SA-BMA shows superior performance in average over five datasets.

| Backbone | Method | IN | IN-V2 | IN-R | IN-A | IN-S | Avg |
|---|---|---|---|---|---|---|---|
| RN50 | Zero-Shot | $59.83_{\pm0.00}$ | $52.89_{\pm0.00}$ | $60.73_{\pm0.00}$ | $\underline{23.25}_{\pm0.00}$ | $35.45_{\pm0.00}$ | $46.43_{\pm0.00}$ |
| | SGD | $61.70_{\pm0.01}$ | $54.31_{\pm0.01}$ | $60.87_{\pm0.01}$ | $22.74_{\pm0.01}$ | $\underline{35.68}_{\pm0.00}$ | $47.06_{\pm0.01}$ |
| | SAM | $61.73_{\pm0.01}$ | $\underline{54.35}_{\pm0.01}$ | $60.86_{\pm0.01}$ | $22.76_{\pm0.01}$ | $35.67_{\pm0.00}$ | $47.07_{\pm0.01}$ |
| | SWAG | $\underline{61.77}_{\pm0.22}$ | $54.10_{\pm0.19}$ | $\mathbf{61.25}_{\pm0.21}$ | $\mathbf{23.25}_{\pm0.08}$ | $35.55_{\pm0.27}$ | $\underline{47.18}_{\pm0.19}$ |
| | SA-BMA | $\mathbf{63.33}_{\pm0.92}$ | $\mathbf{55.06}_{\pm0.79}$ | $\underline{61.14}_{\pm0.37}$ | $22.78_{\pm0.68}$ | $\mathbf{35.82}_{\pm0.11}$ | $\mathbf{47.63}_{\pm0.17}$ |
| ViT-B/16 | Zero-Shot | $68.33_{\pm0.00}$ | $61.91_{\pm0.00}$ | $77.71_{\pm0.00}$ | $49.93_{\pm0.00}$ | $48.22_{\pm0.00}$ | $61.22_{\pm0.00}$ |
| | SGD | $69.97_{\pm0.00}$ | $62.97_{\pm0.01}$ | $78.05_{\pm0.00}$ | $50.31_{\pm0.02}$ | $48.76_{\pm0.00}$ | $62.01_{\pm0.00}$ |
| | SAM | $70.01_{\pm0.02}$ | $63.03_{\pm0.02}$ | $78.03_{\pm0.01}$ | $50.37_{\pm0.00}$ | $48.75_{\pm0.00}$ | $62.04_{\pm0.00}$ |
| | SWAG | $\underline{70.11}_{\pm0.02}$ | $\underline{63.44}_{\pm0.06}$ | $\mathbf{78.33}_{\pm0.03}$ | $\mathbf{50.55}_{\pm0.02}$ | $\underline{48.95}_{\pm0.01}$ | $\underline{62.28}_{\pm0.02}$ |
| | SA-BMA | $\mathbf{72.41}_{\pm0.33}$ | $\mathbf{64.85}_{\pm0.11}$ | $\underline{78.14}_{\pm0.31}$ | $\underline{50.52}_{\pm0.25}$ | $\mathbf{49.25}_{\pm0.03}$ | $\mathbf{63.03}_{\pm0.04}$ |

gizes the advantages of sharpness-aware optimization and Bayesian transfer learning in a few-shot learning context.

In addition, we employ ResNet50 (RN50) and ViT-B/16 in CLIP (Radford et al., 2021), widely-adopted vision-language model (VLM). We fine-tune only the last layer of the CLIP visual encoder on the IN 1K 16-shot dataset and evaluate the trained model on IN and its variants—IN-V2 (Recht et al., 2019), IN-R (Hendrycks et al., 2021a), IN-A (Hendrycks et al., 2021b), and IN-S (Wang et al., 2019)—following the protocols outlined in Radford et al. (2021); Zhu et al. (2023b); Lin et al. (2023). Table 4 shows that SA-BMA outperforms baselines in the in-distribution evaluation and also shows better or comparable robustness in the out-of-distribution datasets both in RN50 and ViT-B/16, which leads to superior performance in average.

## 5.4 ROBUSTNESS ON DISTRIBUTION SHIFT

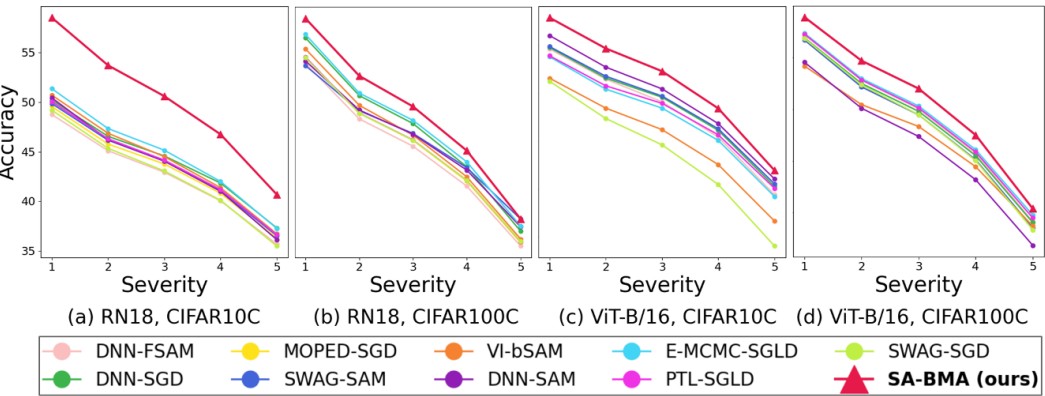

Figure 4: Accuracy under distributional shift. We evaluate the accuracy of RN18 and ViT-B/16 models trained on CIFAR10 and CIFAR100 10-shot across all severity levels of CIFAR10C and CIFAR100C. SA-BMA consistently outperforms all baseline methods across all levels of corruption.

In Figure 4, we show the accuracy on a corrupted dataset, CIFAR10C and CIFAR100C (Hendrycks & Dietterich, 2019), to verify the robustness and generalization performance of SA-BMA. We find that our proposed SA-BMA outperforms other methods across both CIFAR10C and CIFAR100C datasets on backbone models RN18 and ViT-B/16. SA-BMA consistently makes robust predictions across corruption levels from mild level to severe level. We conclude that SA-BMA provides robust predictions under distribution shift across all severities compared to the baselines, as well as under in-distribution image classification. Detailed result is provided in Appendix G.

## 5.5 FLATNESS ANALYSIS

To substantiate the flatness in the loss surface of the SA-BMA model, we compare the sampled models from the posterior approximated with SA-BMA and PTL. The backbone model is RN18, and the trained dataset is CIFAR10 10-shot. As shown in Figure 5, we compare the sampled weight of SA-BMA and PTL in diverse views and show SA-BMA converging to a flatter loss basin with lower loss. Additional results and the protocol to visualize the loss basin are provided in Appendix H.

We also quantitatively compare the sharpness of models. Table 5 presents the results of analyzing the eigenvalue of model Hessian for both DNN and BNN series baseline models, as well as SA-BMA. $\lambda_1$ and $\lambda_5$ represent the largest eigenvalue and the fifth largest eigenvalue, respectively. We used the maximal eigenvalue $\lambda_1$ and ratio $\lambda_1/\lambda_5$ as a metric (Foret et al., 2020; Li et al., 2018; Li & Zhang, 2023). SA-BMA has the lowest value compared to all other baselines, which can be interpreted as our model being the flattest. It supports our visual results, highlighting the superior flatness and improved generalization of SA-BMA.

Table 5: Hessian analysis on RN18 trained with CIFAR10 10-shot. SA-BMA shows the lowest score on both maximal eigenvalue $\lambda_1$ and eigenvalue ratio $\lambda_1/\lambda_5$, proving it leads the model to flatter minima.

| Method | $\lambda_1 \downarrow$ | $\lambda_1/\lambda_5 \downarrow$ |
|---|---|---|
| SGD | 559.62 | 2.59 |
| SAM | 381.74 | 2.23 |
| FSAM | 561.15 | 2.24 |
| bSAM | 532.74 | 2.09 |
| MOPED | 686.90 | 2.41 |
| PTL | 559.16 | 2.23 |
| E-MCMC | 540.83 | 1.98 |
| SWAG | 602.34 | 2.13 |
| F-SWAG | 362.33 | 2.44 |
| SA-BMA | **275.21** | **1.69** |

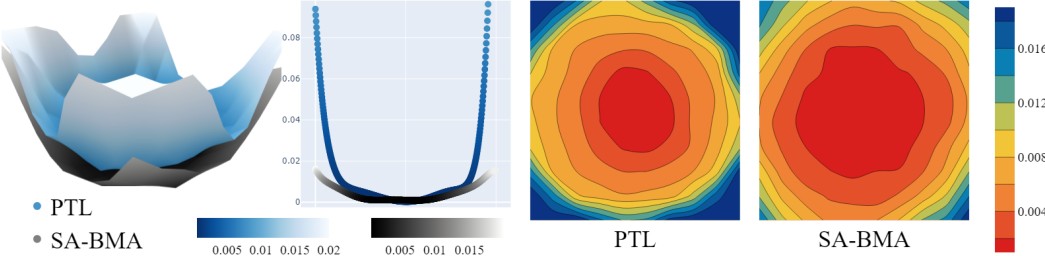

Figure 5: Comparison of the loss surfaces of SA-BMA (grey) and PTL (light blue) models. The loss surface comparison offers an intuitive view of SA-BMA achieving a lower, flatter loss surface than PTL, underlining the importance of flatness in model design.

## 6 RELATED WORKS

### 6.1 FLATNESS AND BNN

Recent works have suggested flat-seeking optimizers combined with BNN. First, SWAG (Maddox et al., 2019) implicitly approximated posterior toward flatter optima based on SWA (Izmailov et al., 2018). However, SWAG can fail to find flat minima, leading to limited improvement in generalization, as shown in Section 3.1. bSAM (Möllenhoff & Khan, 2022) showed that SAM can be interpreted as a relaxation of the Bayes and quantified uncertainty with SAM. Yet, bSAM only focused on uncertainty quantification by simply modifying Adam-based SAM (Khan et al., 2018), not newly considering the parametric geometry for perturbation. Moreover, scaling the variance with the number of data points hampers the direct implementation of bSAM in few-shot settings. SA-BNN (Nguyen et al., 2023) proposed a sharpness-aware posterior derived directly from the variational objective and proved the effectiveness experimentally and theoretically. However, they just employ the L2 norm to calculate the perturbation of SAM without considering the difference between the nature of DNN and BNN. E-MCMC (Li & Zhang, 2023) proposed an efficient MCMC algorithm capable of effectively sampling the posterior within a flat basin by removing the nested chain of Entropy-SGD and Entropy-SGLD. Still, E-MCMC necessitates a guidance model, which doubles the parameters and heavily hinders its employment over large-scale models. SA-BMA is the first to reflect the parameter space in the perturbation step of SAM for stochastic models, considering the nature of BNNs.

### 6.2 BAYESIAN TRANSFER LEARNING

There are several works on performing transfer learning on BNN with prior. PTL (Shwartz-Ziv et al., 2022) constructs BNN by learning closed-form posterior approximation of the pre-trained model on the source task and uses it as a prior for the downstream task after scaling. The work requires additional training on the source task, which makes it restrictive when it is impossible to access the source task dataset. MOPED (Krishnan et al., 2020) employs pre-trained BNN as a prior for VI based on the empirical Bayes method. Using pre-trained DNN, MOPED enhances accessibility to BNN, however, it is only applicable to Mean-field VI (MFVI). Non-parametric transfer learning (NPTL) (Lee et al., 2024) suggested adopting non-parametric learning to make posterior flexible in terms of distribution shift. Our proposed scheme for Bayesian transfer learning can approximate the distribution of parameters within either the downstream dataset or the source dataset, which allows us to leverage more sophisticated and large-scale pre-trained models. Moreover, it is the first study considering flatness in Bayesian transfer learning.

## 7 CONCLUSION

This study shows the limitations of BNNs in capturing the flatness, which is crucial for generalization performance. We also show that BMA can fail to yield optimal results without explicitly considering flatness. To address this issue, we introduce Sharpness-Aware Bayesian Model Averaging (SA-BMA), which seeks to find a flat posterior by capturing flatness in the parameter space. SA-BMA is the generalized version of existing sharpness-aware optimizers for DNN and aligns with the intrinsic nature of BNN. We further propose a Bayesian Transfer Learning scheme, which enables efficient fine-tuning of pre-trained DNNs while maintaining scalability with SA-BMA. Through extensive experiments, we demonstrate that SA-BMA significantly enhances the generalization performance of BNNs in diverse scenarios. Our work highlights the importance of flatness in posterior approximations and provides a practical solution to improve the predictive robustness and accuracy of BNNs.

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

## A ADDITIONAL RESULTS FOR FLATNESS DOSE MATTER FOR BAYESIAN MODEL AVERAGING

### A.1 DETAILS ABOUT HESSIAN EIGENVALUE OF LOSS WITH BMA

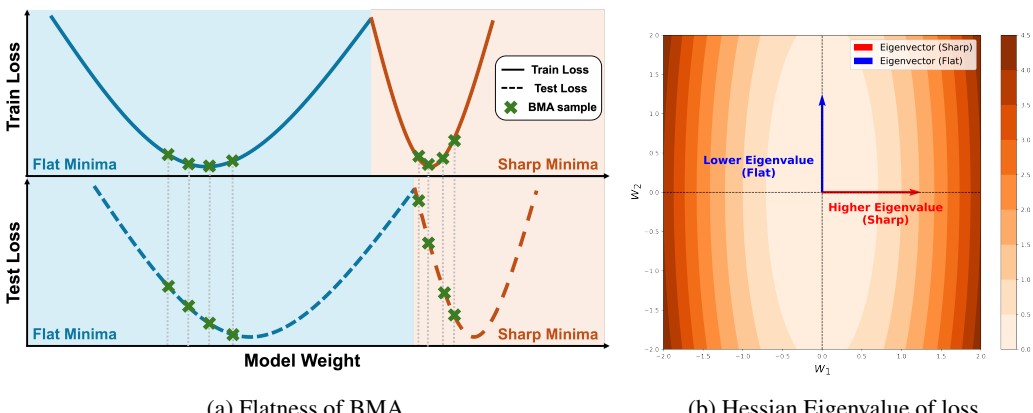

(a) Flatness of BMA

(b) Hessian Eigenvalue of loss

Figure 6: Description of flatness of BMA and Hessian Eigenvalue of loss. (a) depicts how flatness is measured in BNNs. We measure the flatness of individual sampled model weights and subsequently ensemble the flatness of them. (b) represents how the Hessian eigenvalue of loss corresponds to flatness. It reveals that direction of steep curvature (sharp minima) exhibits with larger eigenvalues, while that of gentle curvature (flat minima) exhibits smaller eigenvalues. Based on this understanding, we measure flatness using the maximal eigenvalue of the Hessian at the minima.

To measure the flatness of BNNs and compare them with DNNs, we introduce a new metric specifically designed for this study. Unlike DNNs, where model parameters are typically treated as point estimate, BNNs represent model parameters as random variables, necessitating an appropriate approach for measuring flatness. As shown in Figure 6b, the maximal eigenvalue of the Hessian of the loss function is commonly used to evaluate flatness quantitatively in DNNs (Keskar et al., 2016; Foret et al., 2020; Jastrzebski et al., 2020). To assess flatness in BNNs, we followed BMA protocol. BMA samples model weights from the approximated posterior, calculates the outputs of the sampled individual models, and ensemble the outputs, as shown in Figure 6a. Thus, similar to how BMA operates, we measured the flatness of individual model weights and subsequently ensemble these measurements to derive a comprehensive metric.

## A.2 FLATNESS COMPARISON

To prove that BNN cannot guarantee flatness, we run a variety of experiments for flatness comparison. Since it's uncommon, we do not run DNN or VI with SWAG learning rate scheduler for additional flatness comparisons. First, Table 6 summarizes the comparison results covering different models with a wide range of combinations with learning methods, optimizers, and schedulers. We conducted on RN18 w/o BN and data augmentation. We use MOPED as the VI framework.

Table 6: Flatness Comparison on DNN and BNN with sharpness-aware optimization methods. We observe that BNN does not find the flatness-aware local minima, and the compatibility of BNN and the previous sharpness-aware optimization (SAM) is limited. All experiments were repeated three times with RN18 w/o BN.

| Methods | Dataset Optim | Schedule | CIFAR10 ACC ↑ | ECE ↓ | NLL ↓ | $\lambda_1$ ↓ | $\lambda_1/\lambda_5$ ↓ | CIFAR100 ACC ↑ | ECE ↓ | NLL ↓ | $\lambda_1$ ↓ | $\lambda_1/\lambda_5$ ↓ |
|---|---|---|---|---|---|---|---|---|---|---|---|---|
| DNN | SGD | Constant | $81.96_{\pm0.55}$ | $0.033_{\pm0.004}$ | $0.546_{\pm0.015}$ | $37.34_{\pm3.59}$ | $1.72_{\pm0.22}$ | $44.31_{\pm1.02}$ | $0.047_{\pm0.004}$ | $2.194_{\pm0.032}$ | $12.65_{\pm0.87}$ | $1.68_{\pm0.17}$ |
| | | Cos Decay | $84.51_{\pm0.53}$ | $0.027_{\pm0.015}$ | $0.474_{\pm0.030}$ | $19.98_{\pm2.64}$ | $1.59_{\pm0.09}$ | $47.01_{\pm0.52}$ | $0.074_{\pm0.004}$ | $2.077_{\pm0.027}$ | $102.21_{\pm12.42}$ | $1.73_{\pm0.09}$ |
| | | SWAG lr | $82.72_{\pm3.08}$ | $0.029_{\pm0.023}$ | $0.520_{\pm0.097}$ | $18.09_{\pm6.97}$ | $1.57_{\pm0.06}$ | $33.13_{\pm1.12}$ | $0.065_{\pm0.010}$ | $2.485_{\pm0.362}$ | $26.85_{\pm2.04}$ | $1.90_{\pm0.10}$ |
| | SAM | Constant | $85.06_{\pm0.68}$ | $0.019_{\pm0.006}$ | $0.450_{\pm0.019}$ | $11.81_{\pm0.50}$ | $1.30_{\pm0.04}$ | $51.28_{\pm0.32}$ | $0.043_{\pm0.018}$ | $1.864_{\pm0.014}$ | $6.55_{\pm0.16}$ | $1.45_{\pm0.06}$ |
| | | Cos Decay | $87.29_{\pm0.12}$ | $0.019_{\pm0.003}$ | $0.390_{\pm0.004}$ | $11.71_{\pm0.24}$ | $1.42_{\pm0.01}$ | $54.85_{\pm0.62}$ | $0.031_{\pm0.006}$ | $1.731_{\pm0.007}$ | $6.67_{\pm0.61}$ | $1.43_{\pm0.02}$ |
| | | SWAG lr | $85.34_{\pm2.76}$ | $0.034_{\pm0.015}$ | $0.465_{\pm0.057}$ | $5.57_{\pm0.58}$ | $1.48_{\pm0.05}$ | $48.65_{\pm1.44}$ | $0.053_{\pm0.012}$ | $1.970_{\pm0.065}$ | $7.84_{\pm0.04}$ | $1.45_{\pm0.01}$ |
| SWAG | SGD | Constant | $85.81_{\pm0.42}$ | $0.027_{\pm0.007}$ | $0.426_{\pm0.005}$ | $55.01_{\pm4.74}$ | $1.67_{\pm0.04}$ | $50.35_{\pm0.40}$ | $0.069_{\pm0.031}$ | $2.219_{\pm0.067}$ | $66.59_{\pm20.34}$ | $1.50_{\pm0.04}$ |
| | | Cos Decay | $87.34_{\pm0.48}$ | $0.029_{\pm0.006}$ | $0.384_{\pm0.013}$ | $29.87_{\pm2.55}$ | $1.62_{\pm0.02}$ | $52.31_{\pm0.81}$ | $0.101_{\pm0.065}$ | $2.132_{\pm0.166}$ | $24.58_{\pm15.13}$ | $1.52_{\pm0.06}$ |
| | | SWAG lr | $86.07_{\pm0.12}$ | $0.113_{\pm0.115}$ | $0.435_{\pm0.002}$ | $50.88_{\pm3.00}$ | $1.66_{\pm0.02}$ | $51.52_{\pm0.40}$ | $0.124_{\pm0.008}$ | $2.237_{\pm0.175}$ | $44.21_{\pm5.79}$ | $1.65_{\pm0.01}$ |
| VI | SGD | Constant | $83.11_{\pm0.40}$ | $0.016_{\pm0.002}$ | $0.518_{\pm0.011}$ | $19.72_{\pm0.88}$ | $1.49_{\pm0.04}$ | $45.81_{\pm0.42}$ | $0.018_{\pm0.002}$ | $2.104_{\pm0.017}$ | $217.05_{\pm16.11}$ | $1.67_{\pm0.08}$ |
| | | Cos Decay | $84.51_{\pm0.30}$ | $0.037_{\pm0.003}$ | $0.465_{\pm0.005}$ | $20.86_{\pm0.58}$ | $1.53_{\pm0.01}$ | $48.30_{\pm0.41}$ | $0.121_{\pm0.007}$ | $2.519_{\pm0.029}$ | $34.71_{\pm2.06}$ | $1.80_{\pm0.09}$ |
| MCMC | SGLD | Cos Decay | $86.54_{\pm0.02}$ | $0.043_{\pm0.002}$ | $0.465_{\pm0.005}$ | $24.30_{\pm3.91}$ | $1.83_{\pm0.10}$ | $49.09_{\pm0.45}$ | $0.160_{\pm0.002}$ | $2.777_{\pm0.035}$ | $110.05_{\pm3.19}$ | $1.65_{\pm0.03}$ |

Second, we compare the flatness with ResNet18 pre-trained on ImageNet 1K, provided in *torchvision*. Note that the ResNet18 contains Batch Normalization, and data augmentation is also applied in this setting. Table 7 consistently shows the SWAG cannot guarantee the flatness with Batch Normalization and data augmentation, either.

Table 7: Flatness Comparison on DNN and BNN with sharpness-aware optimization methods. We observe that BNN does not find the flatness-aware local minima even with BN and data augmentation. All experiments were repeated three times with ResNet18 pre-trained on ImageNet 1K.

| Methods | Dataset Optim | Schedule | CIFAR10 ACC ↑ | ECE ↓ | NLL ↓ | $\lambda_1$ ↓ | $\lambda_1/\lambda_5$ ↓ | CIFAR100 ACC ↑ | ECE ↓ | NLL ↓ | $\lambda_1$ ↓ | $\lambda_1/\lambda_5$ ↓ |
|---|---|---|---|---|---|---|---|---|---|---|---|---|
| DNN | SGD | Constant | $95.60_{\pm0.21}$ | $0.005_{\pm0.001}$ | $0.127_{\pm0.002}$ | $587.06_{\pm17.13}$ | $2.22_{\pm0.065}$ | $80.61_{\pm0.17}$ | $0.059_{\pm0.005}$ | $0.674_{\pm0.003}$ | $1261.65_{\pm44.00}$ | $1.97_{\pm0.005}$ |
| | | Cos Decay | $96.46_{\pm0.08}$ | $0.010_{\pm0.001}$ | $0.110_{\pm0.001}$ | $436.79_{\pm22.52}$ | $2.07_{\pm0.276}$ | $82.00_{\pm0.40}$ | $0.058_{\pm0.001}$ | $0.656_{\pm0.007}$ | $1195.30_{\pm33.53}$ | $2.35_{\pm0.086}$ |
| | SAM | Constant | $96.55_{\pm0.11}$ | $0.005_{\pm0.002}$ | $0.102_{\pm0.001}$ | $115.97_{\pm6.09}$ | $1.92_{\pm0.074}$ | $82.06_{\pm0.40}$ | $0.040_{\pm0.004}$ | $0.617_{\pm0.005}$ | $306.44_{\pm43.19}$ | $2.19_{\pm0.190}$ |
| | | Cos Decay | $96.92_{\pm0.14}$ | $0.031_{\pm0.002}$ | $0.112_{\pm0.002}$ | $728.99_{\pm5.07}$ | $2.06_{\pm0.053}$ | $83.76_{\pm0.13}$ | $0.060_{\pm0.001}$ | $0.589_{\pm0.003}$ | $382.66_{\pm10.00}$ | $2.07_{\pm0.052}$ |
| SWAG | SGD | Constant | $96.46_{\pm0.09}$ | $0.084_{\pm0.026}$ | $0.178_{\pm0.031}$ | $2263.44_{\pm1025.76}$ | $2.86_{\pm0.886}$ | $82.99_{\pm0.20}$ | $0.118_{\pm0.016}$ | $0.677_{\pm0.022}$ | $2227.27_{\pm539.94}$ | $3.00_{\pm0.510}$ |
| | | Cos Decay | $96.54_{\pm0.06}$ | $0.021_{\pm0.001}$ | $0.116_{\pm0.002}$ | $1147.92_{\pm165.90}$ | $2.74_{\pm0.357}$ | $82.48_{\pm0.22}$ | $0.083_{\pm0.001}$ | $0.667_{\pm0.006}$ | $2457.46_{\pm224.32}$ | $2.93_{\pm0.086}$ |
| | | SWAG lr | $96.35_{\pm0.10}$ | $0.090_{\pm0.011}$ | $0.192_{\pm0.015}$ | $7823.13_{\pm2183.11}$ | $3.03_{\pm0.362}$ | $82.34_{\pm0.19}$ | $0.052_{\pm0.002}$ | $0.638_{\pm0.003}$ | $1529.73_{\pm58.99}$ | $3.02_{\pm0.102}$ |

Third, we train the pre-trained RN18 with CIFAR10 10-shot and CIFAR100 10-shot. In other words, we only use 10 data per class in this setting. Table 8 shows identical results with other flatness comparisons. SWAG cannot guarantee flatness without SAM.

Table 8: Flatness Comparison on DNN and BNN with sharpness-aware optimization method. We observe that BNN does not find the flatness-aware local minima, and the compatibility of BNN and the previous sharpness-aware optimization (SAM) is limited. All experiments were repeated three times with ResNet18 pre-trained ImageNet 1K in few-shot setting.

| Methods | Dataset Optim | Schedule | CIFAR10 10-shot ACC ↑ | ECE ↓ | NLL ↓ | $\lambda_1$ ↓ | $\lambda_1/\lambda_5$ ↓ | CIFAR100 10-shot ACC ↑ | ECE ↓ | NLL ↓ | $\lambda_1$ ↓ | $\lambda_1/\lambda_5$ ↓ |
|---|---|---|---|---|---|---|---|---|---|---|---|---|
| DNN | SGD | Constant | $81.96_{\pm0.55}$ | $0.033_{\pm0.015}$ | $0.546_{\pm0.015}$ | $37.34_{\pm3.59}$ | $1.73_{\pm0.229}$ | $44.31_{\pm1.02}$ | $0.047_{\pm0.229}$ | $2.194_{\pm0.032}$ | $12.65_{\pm0.87}$ | $1.68_{\pm0.172}$ |
| | | Cos Decay | $84.51_{\pm0.53}$ | $0.027_{\pm0.015}$ | $0.474_{\pm0.030}$ | $19.98_{\pm2.64}$ | $1.59_{\pm0.097}$ | $47.01_{\pm0.52}$ | $0.074_{\pm0.004}$ | $2.077_{\pm0.027}$ | $102.21_{\pm12.42}$ | $1.74_{\pm0.097}$ |
| | SAM | Constant | $57.91_{\pm1.84}$ | $0.030_{\pm0.017}$ | $1.227_{\pm0.038}$ | $199.79_{\pm47.69}$ | $2.25_{\pm0.426}$ | $45.54_{\pm0.64}$ | $0.094_{\pm0.011}$ | $2.118_{\pm0.021}$ | $359.59_{\pm38.51}$ | $1.86_{\pm0.049}$ |
| | | Cos Decay | $56.54_{\pm2.57}$ | $0.015_{\pm0.005}$ | $1.255_{\pm0.070}$ | $148.59_{\pm17.70}$ | $2.12_{\pm0.190}$ | $45.51_{\pm1.26}$ | $0.106_{\pm0.009}$ | $2.131_{\pm0.051}$ | $426.73_{\pm51.64}$ | $2.08_{\pm0.293}$ |
| SWAG | SGD | Constant | $55.74_{\pm1.57}$ | $0.018_{\pm0.002}$ | $1.289_{\pm0.047}$ | $530.71_{\pm48.39}$ | $1.53_{\pm0.314}$ | $44.22_{\pm1.27}$ | $0.120_{\pm0.003}$ | $2.221_{\pm0.069}$ | $976.07_{\pm178.03}$ | $2.42_{\pm0.276}$ |
| | | Cos Decay | $56.47_{\pm0.77}$ | $0.042_{\pm0.013}$ | $1.274_{\pm0.036}$ | $566.26_{\pm96.65}$ | $2.18_{\pm0.329}$ | $44.18_{\pm1.26}$ | $0.102_{\pm0.009}$ | $2.225_{\pm0.051}$ | $588.19_{\pm70.21}$ | $2.46_{\pm0.293}$ |
| | | SWAG lr | $56.13_{\pm1.37}$ | $0.041_{\pm0.007}$ | $1.272_{\pm0.059}$ | $562.87_{\pm116.08}$ | $2.16_{\pm0.373}$ | $43.91_{\pm1.00}$ | $0.106_{\pm0.004}$ | $2.214_{\pm0.058}$ | $756.86_{\pm68.28}$ | $2.10_{\pm0.160}$ |

Fourth, we measure the flatness of last-layer SWAG (L-SWAG) and VI (L-VI) in Table 9. We use the trained DNN models as an initial weight for L-SWAG. For example, a DNN model trained with constant lr scheduling and SGD optimizer is the base model for L-SWAG, which trains with the same lr schedule and optimizer. For L-VI, we only set stochastic parameters for last layer. Again, SWAG failed to show better flatness compared to DNN with SAM.

Table 9: Flatness Comparison on DNN and L-SWAG applied to the DNN. We confirm that using pre-existed flatness-aware optimization on the last layer of BNN cannot be enough to pull the model to a flat basin. All experiments were repeated three times with ResNet18 w/o BN.

| Methods | Dataset Optim | Schedule | CIFAR10 ACC ↑ | ECE ↓ | NLL ↓ | $\lambda_1$ ↓ | $\lambda_1/\lambda_5$ ↓ | CIFAR100 ACC ↑ | ECE ↓ | NLL ↓ | $\lambda_1$ ↓ | $\lambda_1/\lambda_5$ ↓ |
|---|---|---|---|---|---|---|---|---|---|---|---|---|
| DNN | SGD | Constant | $81.96_{\pm0.55}$ | $0.033_{\pm0.004}$ | $0.546_{\pm0.015}$ | $37.34_{\pm3.59}$ | $1.73_{\pm0.229}$ | $44.31_{\pm1.02}$ | $0.047_{\pm0.004}$ | $2.194_{\pm0.032}$ | $12.65_{\pm0.87}$ | $1.68_{\pm0.172}$ |
| | | Cos Decay | $84.51_{\pm0.53}$ | $0.027_{\pm0.015}$ | $0.474_{\pm0.030}$ | $19.98_{\pm2.64}$ | $1.59_{\pm0.097}$ | $47.01_{\pm0.52}$ | $0.074_{\pm0.004}$ | $2.077_{\pm0.027}$ | $102.21_{\pm12.42}$ | $1.74_{\pm0.097}$ |
| | | SWAG lr | $82.72_{\pm3.08}$ | $0.029_{\pm0.023}$ | $0.520_{\pm0.097}$ | $18.09_{\pm6.97}$ | $1.57_{\pm0.068}$ | $33.13_{\pm1.12}$ | $0.065_{\pm0.010}$ | $2.485_{\pm0.362}$ | $26.85_{\pm2.04}$ | $1.90_{\pm0.107}$ |
| | SAM | Constant | $85.06_{\pm0.68}$ | $0.019_{\pm0.006}$ | $0.450_{\pm0.019}$ | $11.81_{\pm0.50}$ | $1.31_{\pm0.020}$ | $51.28_{\pm0.32}$ | $0.043_{\pm0.018}$ | $1.864_{\pm0.014}$ | $6.55_{\pm0.16}$ | $1.45_{\pm0.061}$ |
| | | Cos Decay | $87.29_{\pm0.12}$ | $0.019_{\pm0.003}$ | $0.390_{\pm0.004}$ | $11.71_{\pm0.24}$ | $1.43_{\pm0.006}$ | $54.85_{\pm0.62}$ | $0.031_{\pm0.006}$ | $1.731_{\pm0.007}$ | $6.67_{\pm0.61}$ | $1.44_{\pm0.026}$ |
| | | SWAG lr | $85.34_{\pm2.76}$ | $0.034_{\pm0.015}$ | $0.465_{\pm0.057}$ | $5.57_{\pm0.58}$ | $1.49_{\pm0.053}$ | $48.65_{\pm1.44}$ | $0.053_{\pm0.012}$ | $1.970_{\pm0.065}$ | $7.84_{\pm0.04}$ | $1.45_{\pm0.013}$ |
| L-SWAG | SGD | Constant | $82.20_{\pm0.40}$ | $0.036_{\pm0.007}$ | $0.544_{\pm0.007}$ | $49.49_{\pm4.71}$ | $1.68_{\pm0.080}$ | $49.65_{\pm0.65}$ | $0.053_{\pm0.003}$ | $1.980_{\pm0.012}$ | $18.65_{\pm1.14}$ | $1.71_{\pm0.134}$ |
| | | Cos Decay | $84.94_{\pm0.70}$ | $0.034_{\pm0.01}$ | $0.475_{\pm0.012}$ | $26.26_{\pm2.81}$ | $1.62_{\pm0.055}$ | $49.56_{\pm0.58}$ | $0.015_{\pm0.003}$ | $1.938_{\pm0.014}$ | $131.32_{\pm13.82}$ | $1.65_{\pm0.046}$ |
| | | SWAG lr | $83.26_{\pm3.10}$ | $0.043_{\pm0.027}$ | $0.544_{\pm0.052}$ | $49.49_{\pm8.86}$ | $1.68_{\pm0.134}$ | $49.65_{\pm0.65}$ | $0.053_{\pm0.002}$ | $1.980_{\pm0.036}$ | $18.65_{\pm3.26}$ | $1.71_{\pm0.021}$ |
| L-VI | SGD | Constant | $83.46_{\pm0.34}$ | $0.027_{\pm0.002}$ | $0.529_{\pm0.000}$ | $29.65_{\pm3.44}$ | $1.64_{\pm0.087}$ | $49.26_{\pm0.69}$ | $0.046_{\pm0.006}$ | $2.068_{\pm0.018}$ | $63.17_{\pm7.96}$ | $1.61_{\pm0.165}$ |
| | | Cos Decay | $85.11_{\pm0.35}$ | $0.083_{\pm0.003}$ | $0.605_{\pm0.007}$ | $55.92_{\pm3.02}$ | $1.56_{\pm0.076}$ | $50.70_{\pm2.29}$ | $0.130_{\pm0.083}$ | $2.424_{\pm0.337}$ | $47.53_{\pm38.46}$ | $1.61_{\pm0.24}$ |

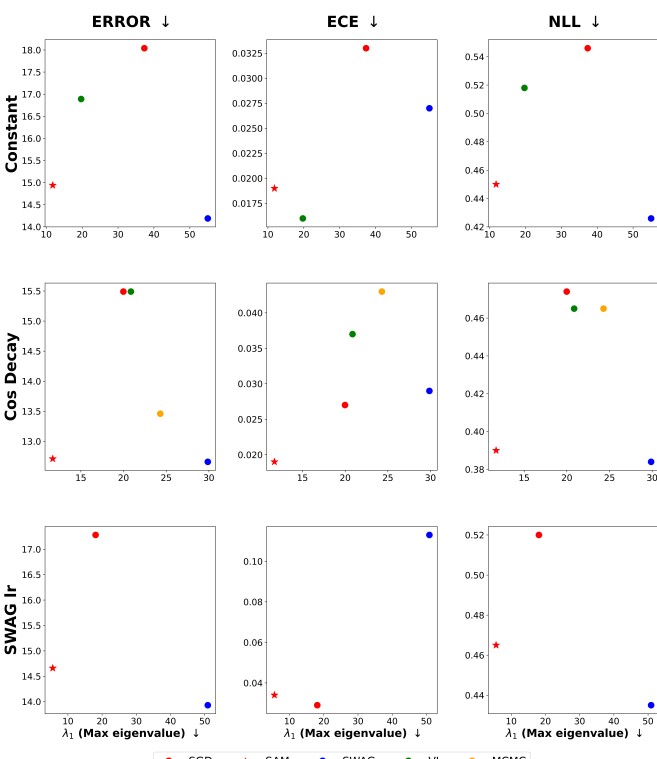

Figure 7: Comparison of Error, NLL, and ECE with various schedulers on CIFAR10 in relation to the maximum eigenvalue $\lambda_1$.

In Figure 1, only the Error and NLL in relation to the maximum eigenvalue $\lambda_1$ are presented for training with the Constant scheduler. Figures 7 and 8 show the Error, NLL, and additionally ECE with various schedulers and datasets. These figures depict the experimental results on CIFAR10 and CIFAR100, respectively. Similar to what was observed in Figure 1, BNN does not guarantee the flatness.

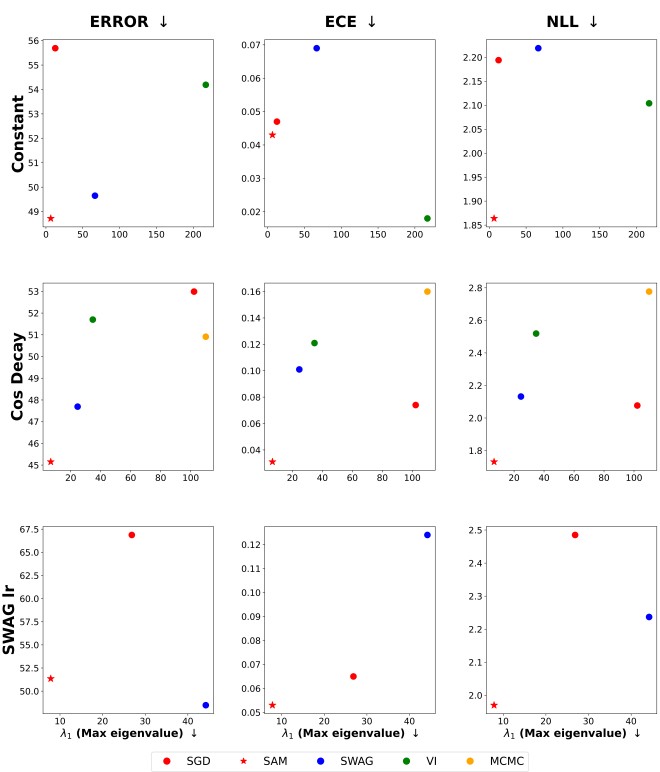

Figure 8: Comparison of Error, NLL, and ECE with various schedulers on CIFAR100 in relation to the maximum eigenvalue $\lambda_1$.

## A.3 CORRELATION BETWEEN FLATNESS AND GENERALIZATION

Together with flatness comparisons in A.2, we check the correlation between flatness and generalization performance of sampled models throughout all considered learning rate schedulers. We present the scatter plot of the model, sampled from RN18 w/o BN trained on CIFAR10 and CIFAR100 in the first and second rows of Figure 9. Each column of Figure 9 denotes Constant scheduler, Cosine Decay scheduler, and SWAG lr scheduler, respectively. All the models are trained with SWAG and SGD momentum, and we set maximal eigenvalue $\lambda_1$ as a flatness measure. Correlation with flatness and each generalization performance metric is suggested in the legend, as well. Regardless of the scheduler and dataset, all generalization performances, error, ECE, and NLL strongly correlate with flatness.

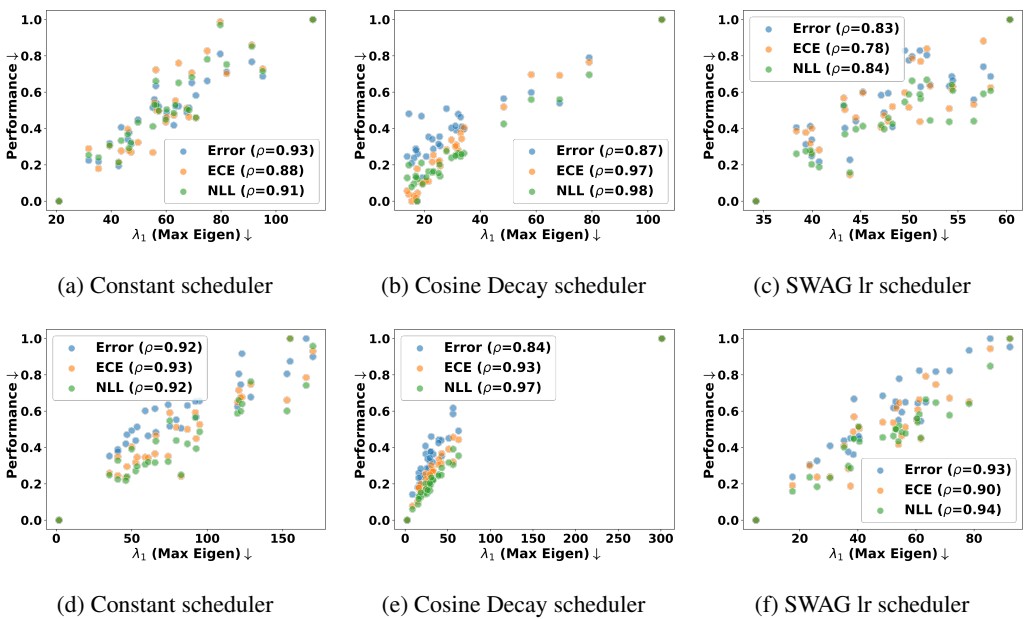

(a) Constant scheduler      (b) Cosine Decay scheduler      (c) SWAG lr scheduler

(d) Constant scheduler      (e) Cosine Decay scheduler      (f) SWAG lr scheduler

Figure 9: Correlation between maximal eigenvalue and performances of 30 sampled models from SWAG throughout all considered schedulers. It shows classification error, ECE, and NLL are distinctly correlated with flatness. We conjecture that the flatness is crucial for the generalization performance of BNN

## A.4 PROGRESSIVE BMA BASED ON FLATNESS

We also inspect the influence of flatness on BMA performance throughout all considered schedulers. We prepared 30 sampled models, trained on CIFAR10 and CIFAR100 with RN18 w/o BN. "Flat" denotes starting BMA from sampling the flattest model. "Sharp" denotes starting BMA by sampling the sharpest model. "Rand" denotes starting BMA from a random sample of prepared 30 models. Figure 10 and 11 shows the results in CIFAR10 and CIFAR100, respectively. Each row means Constant, Cosine Decay, and SWAG lr scheduler, and each column denotes the classification error, ECE, and NLL.

Generally, we observe that there is no significant improvement or limited improvement in performance when gradually applying BMA from flat models. Following the observation, we conclude flatness should be taken into account for efficient BMA. This observation is particularly pronounced in classification error ((a), (d), (g) of Figure 10 and 11). However, the trend is inconsistent in ECE. It is an interesting topic for future research.

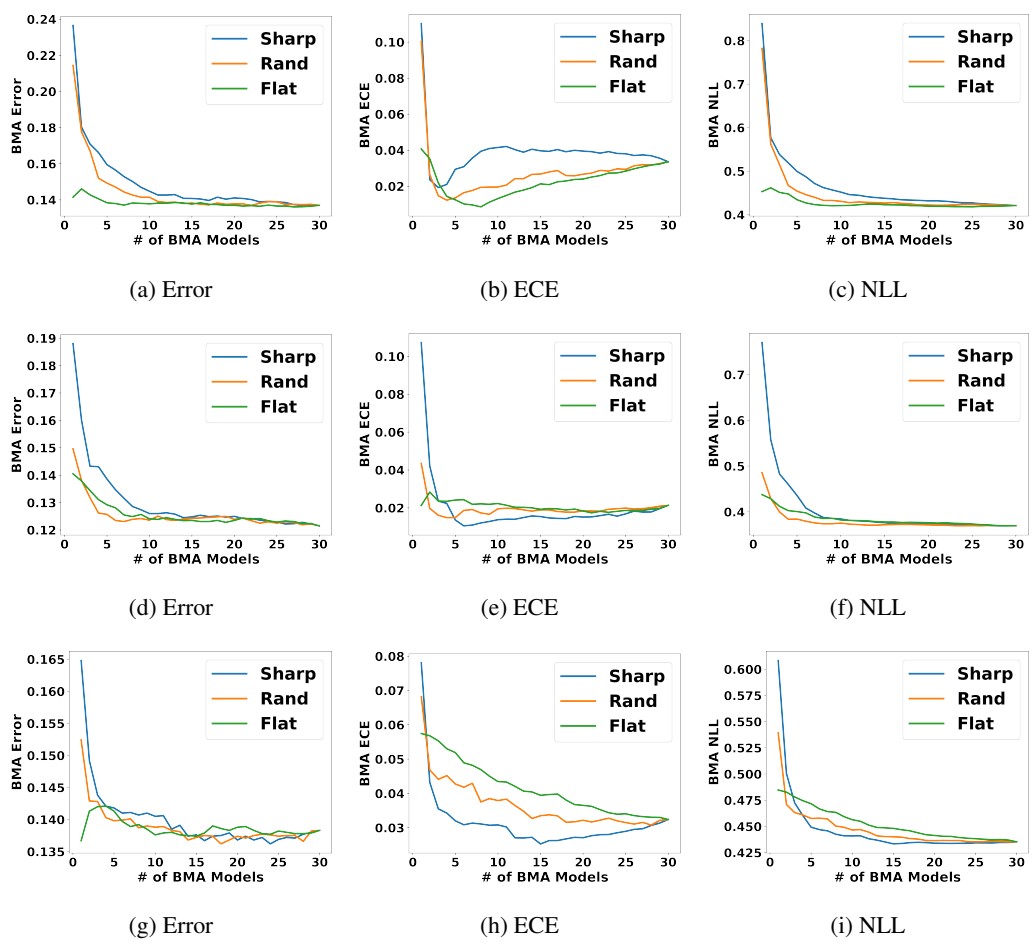

Figure 10: Performance variation based on sampling considering flatness among BMA on CIFAR10. Each row means the Constant, Cos Decay, and SWAG lr scheduler. Each column denotes classification error, ECE, and NLL. "Flat" denotes starting BMA from sampling the flattest model, and "Sharp" means the opposite of "Flat". "Rand" denotes starting BMA from a random sample of prepared 30 models. It reveals that the flatness should be taken into account for efficient BMA.

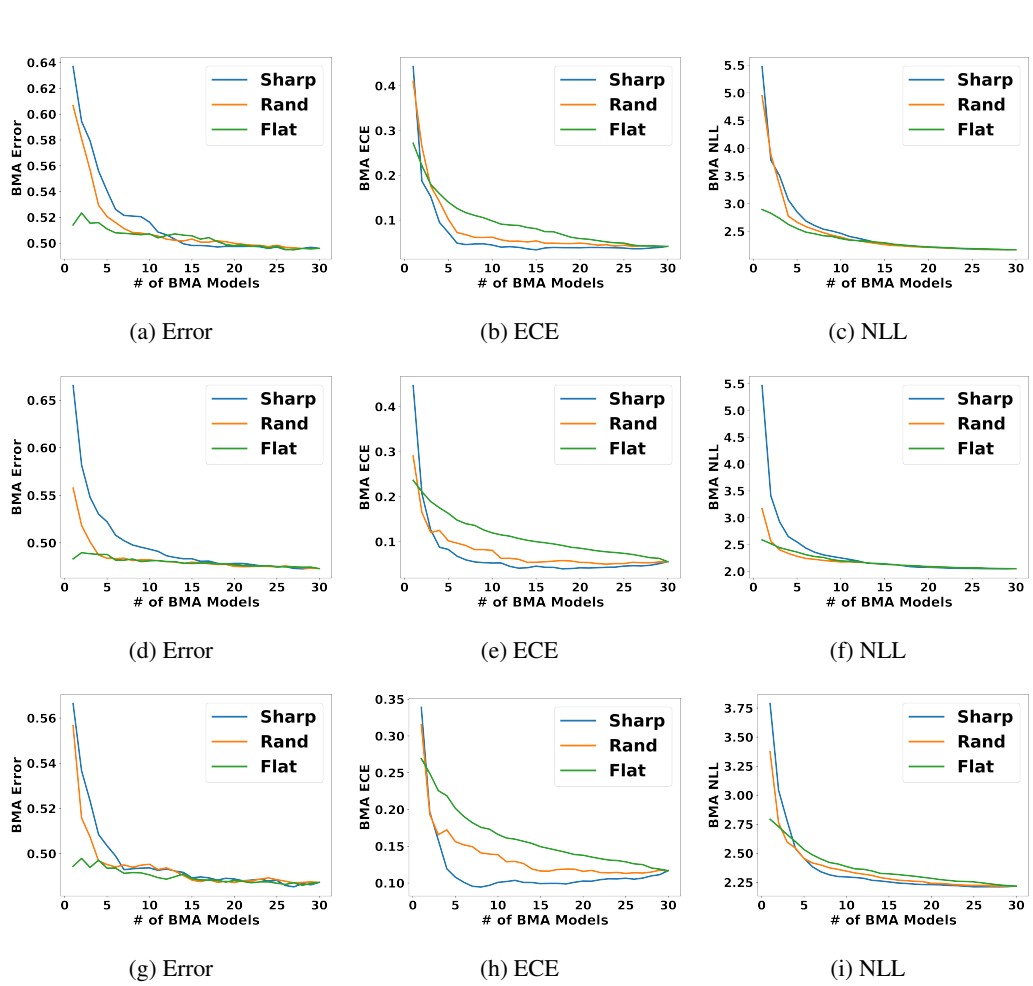

(a) Error          (b) ECE          (c) NLL

(d) Error          (e) ECE          (f) NLL

(g) Error          (h) ECE          (i) NLL

Figure 11: Performance variation based on sampling considering flatness among BMA on CIFAR100. Each row means the Constant, Cos Decay, and SWAG lr scheduler. Each column denotes classification error, ECE, and NLL. "Flat" denotes starting BMA from sampling the flattest model, and "Sharp" means the opposite of "Flat". "Rand" denotes starting BMA from a random sample of prepared 30 models. It reveals that the flatness should be taken into account for efficient BMA.

## B ADDITIONAL RESULTS FOR SYNTHETIC EXAMPLE

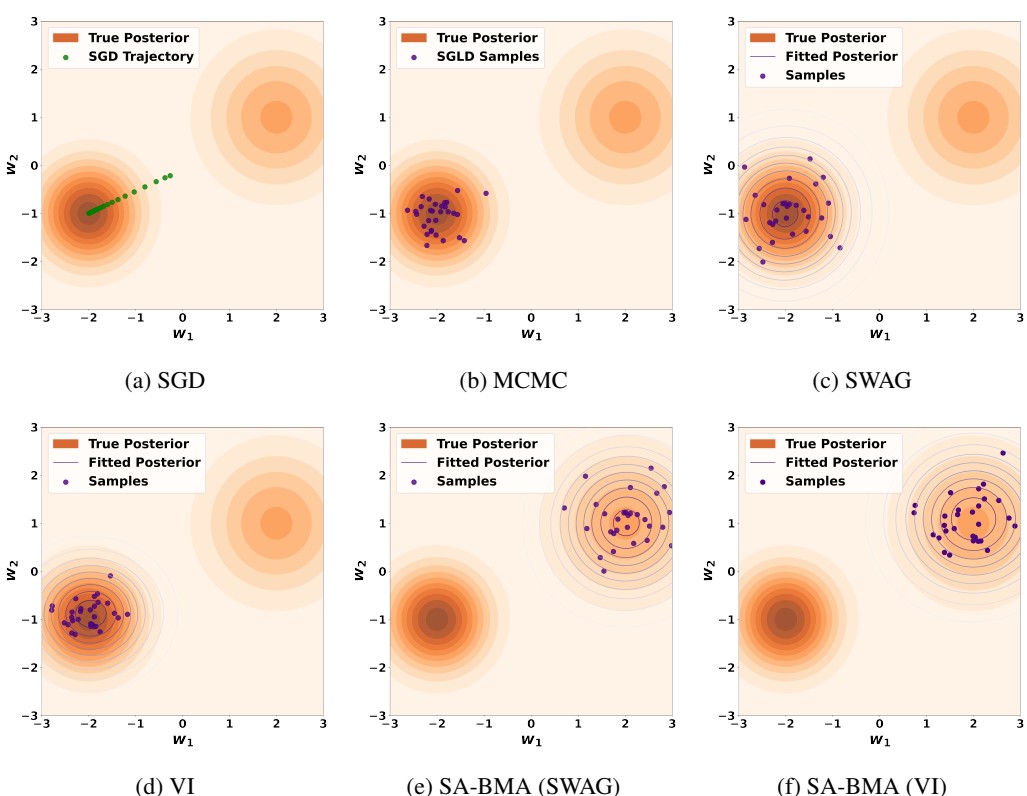

(a) SGD      (b) MCMC      (c) SWAG

(d) VI      (e) SA-BMA (SWAG)      (f) SA-BMA (VI)

Figure 12: Posterior approximation with synthetic example. When both flat and sharp modes coexist, we compared how optimizers approximate the posterior. Unlike other methods, the proposed SA-BMA converged to the flat mode, demonstrating its effectiveness in finding more stable solutions.

Following Li & Zhang (2023), we construct a loss surface following the distribution $\frac{1}{2}(\mathcal{N}([-2,-1]^T, 0.5I)) + \frac{1}{2}(\mathcal{N}([2,1]^T, I))$ and set the initial point at $(-0.4, -0.4)$. Unlike other SGD-based methods, SA-BMA efficiently identifies flat modes regardless of the underlying BNN frameworks.

## C    EXPERIMENTAL DETAILS: LEARNING FROM SCRATCH

### C.1    SA-BMA WITH DIVERSE BNN FRAMEWORKS

In Eq. (4), SA-BMA can be applied with various BNN frameworks by using an empirical loss function $l(\cdot)$ and adjusting the parameter $\beta$. We commonly set $l(\cdot)$ as cross-entropy loss in context of image classification task. Note that SA-BMA was applied only to the normalization layers and the last layer, while all other layers were trained using SGD.

**SA-BMA (VI)**    For VI, we follow the loss function of Eq. (4).

**SA-BMA (MCMC)**    We mainly adopt SGLD for MCMC in this work. For SGLD, we incorporated noise into Eq. (4) without KLD term ($\beta = 0$) based on the learning rate and the hyperparameter, temperature. In this approach, during the first step, the adversarial posterior is computed without any noise (Eq. (7)). In the second step, both the noise and the adversarial posterior are used together in the learning process.

**SA-BMA (SWAG)**    SWAG updates the first and second moments along the trajectory of SWA and uses these moments to approximate the posterior with a Gaussian distribution. In Eq. (4), $\beta$ is fixed to 0, and as the trajectory of SWA is optimized through SA-BMA, posterior approximation can be performed accordingly.

### C.2    HYPERPARAMETERS FOR EXPERIMENTS

In this section, we provide the details of the experimental setup for Section 5.2. In the other experiments, the range of hyperparameters, excluding the number of epochs, is shared across different backbones and methods. For all experiments, the hyperparameters are selected using grid-search. Configuration of best hyperparameters for each baseline is summarized in Table 10 and Table 11.

Table 10: Hyperparameter Configuration for CIFAR10

| Backbone | Baseline | learning rate | $\beta_1$ (momentum) | $\beta_2$ | $\gamma$ | weight decay |
|---|---|---|---|---|---|---|
| RN18 | SGD | 5e-2 | 9e-1 | $\times$ | $\times$ | 5e-4 |
| | SAM | 1e-1 | 9e-1 | $\times$ | 1e-1 | 5e-4 |
| | FSAM | 5e-2 | 9e-1 | $\times$ | 1e-2 | 5e-4 |
| | bSAM | 8e-1 | 9e-1 | 0.999 | 1e-1 | 5e-4 |
| | VI | 5e-3 | 9e-1 | $\times$ | $\times$ | 5e-4 |
| | **SA-BMA (VI)** | 5e-2 | 9e-1 | $\times$ | 1e-1 | 5e-4 |
| | MCMC | 1e-1 | $\times$ | $\times$ | $\times$ | 5e-4 |
| | E-MCMC | 1e-1 | $\times$ | $\times$ | $\times$ | 5e-4 |
| | **SA-BMA (MCMC)** | 5e-2 | 9e-1 | $\times$ | 5e-2 | 5e-4 |
| | SWAG | 1e-1 | 9e-1 | $\times$ | $\times$ | 5e-4 |
| | F-SWAG | 1e-1 | 9e-1 | $\times$ | 1e-1 | 5e-4 |
| | **SA-BMA (SWAG)** | 1e-1 | 9e-1 | $\times$ | 1e-1 | 5e-4 |
| ViT-B/16[†] | SGD | 1e-1 | 9e-1 | $\times$ | $\times$ | 5e-4 |
| | SAM | 1e-1 | 9e-1 | $\times$ | 5e-2 | 5e-4 |
| | FSAM | 1e-1 | 9e-1 | $\times$ | 1e-1 | 5e-4 |
| | bSAM | 5e-1 | 9e-1 | 0.999 | 1e-1 | 5e-4 |
| | VI | 5e-3 | 9e-1 | $\times$ | $\times$ | 5e-4 |
| | **SA-BMA (VI)** | 5e-3 | 9e-1 | $\times$ | 5e-3 | 5e-4 |
| | MCMC | 2e-2 | $\times$ | $\times$ | $\times$ | 5e-4 |
| | EMCMC | 2e-2 | $\times$ | $\times$ | $\times$ | 5e-4 |
| | **SA-BMA (MCMC)** | 3e-2 | 9e-1 | $\times$ | 1e-2 | 5e-4 |
| | SWAG | 5e-2 | 9e-1 | $\times$ | $\times$ | 5e-4 |
| | F-SWAG | 5e-2 | 9e-1 | $\times$ | | 5e-4 |
| | **SA-BMA (SWAG)** | 5e-2 | 9e-1 | $\times$ | 1e-2 | 5e-4 |

Table 11: Hyperparameter Configuration for CIFAR100

| Backbone | Baseline | learning rate | $\beta_1$ (momentum) | $\beta_2$ | $\gamma$ | weight decay |
|---|---|---|---|---|---|---|
| RN18 | SGD | 1e-1 | 9e-1 | $\times$ | $\times$ | 5e-4 |
| | SAM | 5e-2 | 9e-1 | $\times$ | 1e-1 | 5e-4 |
| | FSAM | 1e-1 | 9e-1 | $\times$ | 1e-2 | 5e-4 |
| | bSAM | 1 | 9e-1 | 0.999 | 1e-1 | 5e-4 |
| | VI | 5e-3 | 9e-1 | $\times$ | $\times$ | 5e-4 |
| | **SA-BMA (VI)** | 8e-3 | 9e-1 | $\times$ | 2e-1 | 5e-4 |
| | MCMC | 5e-1 | $\times$ | $\times$ | $\times$ | 5e-4 |
| | E-MCMC | 5e-1 | $\times$ | $\times$ | $\times$ | 5e-4 |
| | **SA-BMA (MCMC)** | 1e-1 | 9e-1 | $\times$ | 3e-2 | 5e-4 |
| | SWAG | 1e-1 | 9e-1 | $\times$ | $\times$ | 5e-4 |
| | F-SWAG | 1e-1 | 9e-1 | $\times$ | 1e-1 | 5e-4 |
| | **SA-BMA (SWAG)** | 3e-1 | 9e-1 | $\times$ | 2e-1 | 5e-4 |
| ViT-B/16$^\dagger$ | SGD | 1e-1 | 9e-1 | $\times$ | $\times$ | 5e-4 |
| | SAM | 1e-1 | 9e-1 | $\times$ | 1e-1 | 5e-4 |
| | FSAM | 1e-1 | 9e-1 | $\times$ | 1e-2 | 5e-4 |
| | bSAM | 5e-1 | 9e-1 | 0.999 | 1e-1 | 5e-4 |
| | VI | 3e-2 | 9e-1 | $\times$ | $\times$ | 5e-4 |
| | **SA-BMA (VI)** | 8e-3 | 9e-1 | $\times$ | 1e-1 | 5e-4 |
| | MCMC | 2e-1 | $\times$ | $\times$ | $\times$ | 5e-4 |
| | EMCMC | 1e-1 | $\times$ | $\times$ | $\times$ | 5e-4 |
| | **SA-BMA (MCMC)** | 5e-2 | 9e-1 | $\times$ | 5e-2 | 5e-4 |
| | SWAG | 1e-1 | 9e-1 | $\times$ | $\times$ | 5e-4 |
| | F-SWAG | 1e-1 | 9e-1 | $\times$ | 1e-1 | 5e-4 |
| | **SA-BMA (SWAG)** | 1e-1 | 9e-1 | $\times$ | 1e-1 | 5e-4 |

**Stochastic Gradient Descent with Momentum (SGD)**    In this study, we adopt Stochastic Gradient Descent with Momentum as an optimizer for DNN. Learning rate schedule is fixed to cosine decay. We run 300 epochs. The hyperparameter tuning range included learning rate in [1e-4, 1e-3, 1e-2].

**Sharpness Aware Minimization (SAM)**    We set SGD with momentum as the base optimizer of SAM. It also ran upon a cosine decay learning rate scheduler. All the range of hyperparameters is shared with SGD with Momenmtum. Additional hyperparameter $\gamma$, the ball size of perturbation, is in [1e-2, 5e-2, 0.1].

**Fisher SAM (FSAM)**    We set SGD with momentum as the base optimizer of FSAM. It also ran upon a cosine decay learning rate scheduler. All the range of hyperparameters is shared with SGD with Momenmtum. Additional hyperparameter $\eta$, regularize Fisher impact, is in [1e-2, 1e-1, 1].

**SAM as an optimal relaxation of Bayes (bSAM)**    We use a cosine learning rate decay scheme. We run 300 epochs with fixed $\beta_1$ and $\beta_2$. The hyperparameter tuning rage included: learning rate in [1e-1, 3e-1, 5e-1, 8e-1, 1], weight decay in [1e-4, 5e-4, 1e-3, 1e-2], damping in [1e-1, 1e-2, 1e-3], and $\gamma$ in [1e-3, 1e-2, 5e-2, 1e-1, 5e-1]. Damping parameter stabilizes the method by adding constant when updating variance estimate.

**Variational Inference (VI)**    We use MOPED to change DNN into BNN, first. We set prior mean and variance as 0 and 1, respectively. Besides, we set the posterior mean as 0 and variance as 1e-3. We adopt Reparameterization as type of VI. The essential hyperparmeter for MOPED is $\delta$, which adjusts how much to incorporate pre-trained weights. The $\delta$ was searched in [1e-3, 5e-3, 1e-2]. Moreover, we add a hyperparameter $\beta$ for MOPED that can balance the loss term in VI. The $\beta$ is in range [1e-2, 1e-1 ,1]

**MCMC**    We consistently use SGLD (Welling & Teh, 2011) for MCMC in this work. It ran upon a cyclic cosine decay learning rate scheduler. The number of cycles was ranged in [2, 4]. The number of sampled models is in [10, 20, 28]. We search temperature in [1e-5, 5e-4, 1e-4, 5e-3, 1e-3, 1e-2].

**Entropy-MCMC (E-MCMC)**    We use a cosine learning rate decay scheme, annealing the learning rate to zero. We run 300 epochs. We search $\eta$ in [1e-4, 5e-3, 1e-3, 5e-2, 1e-2, 1e-1] and a system temperature $T$ in [1e-4, 5e-4, 1e-3, 5e-3, 1e-2]. Note that the $\eta$ handles flatness, and the system temperature adjusts the weight update's step size.

**SWAG**    We use a cosine learning rate decay scheme for SWAG. All the range of hyperparameters is shared with SGD with Momenmtum. Additionally, we search for three additional hyperparameters for SWAG, capturing DNN snapshots and calculating statistics. First, the epoch to start SWA is in [161, 201], and epoch is 300. Second, the frequency of capturing the model snapshot is in [1, 2, 3]. Third, the low rank for covariance is in [2, 3, 5, 7, 10].

**F-SWAG**    F-SWAG shares hyperparameter with SWAG, except $\gamma$. We search $\gamma$ in [1e-2, 5e-2, 1e-1].

**Sharpness-aware Bayesian Model Averaging (SA-BMA)**    In case of SA-BMA (VI), we set $\mathcal{N}(0, 1e-3)$ as prior and $\delta$ as 1e-3 to make DNN to BNN using MOPED. After getting prior distribution, we search three hyperparameters: learning rate and $\gamma$. The hyperparameter tuning range included: learning rate in [1e-3, 5e-3, 1e-2, 5e-2], $\gamma$ in [1e-2, 5e-2, 1e-1, 5e-1]. We set weight decay as $5e-4$ for all backbones and train the model over 300 epochs with early stopping. We fix $\beta$ as 1e-8 for all experiments. In case of SA-BMA (MCMC), we search learning rate, temperature for learning rate scheduling, and $\gamma$. The hyperparameter ranges are [1e-3, 5e-3, 1e-2, 5e-2] for learning rate, [1e-4, 5e-3, 1e-3, 5e-2, 1e-2, 1e-1] for temperature, and [5e-3, 1e-2, 5e-2, 1e-1, 5e-1] for $\gamma$. In case of SA-BMA (SWAG), we follow the hyperparameter for SWAG, except $\gamma$ in [1e-2, 5e-2, 1e-1].

# D    EXPERIMENTAL DETAILS:FEW-SHOT IMAGE CLASSIFICATION WITH BAYESIAN TRANSFER LEARNING

## D.1    SA-BMA WITH DIVERSE BNN FRAMEWORKS

Diverse BNN frameworks can be adopted for Bayesian Transfer Learning. Specifically, there are several options for making pre-trained DNN into BNN. In this work, we mainly adopt MOPED and SWAG for the converting.

In addition, SA-BMA can be applied with various BNN frameworks by using an empirical loss function $l(\cdot)$ and adjusting the parameter $\beta$ in Eq. (9). We commonly set $l(\cdot)$ as cross-entropy loss in context of image classification task.

**SA-BMA (VI)**    First, we convert pre-trained DNN into BNN with MOPED. We set the converted BNN as prior, $q_\theta^{\mathrm{pr}}(w|\mathcal{D}^{\mathrm{pr}})$ in Eq. (9), and initial point of model. We only train parameters of normalization and last layer and freeze others. We train them with the loss function of Eq. (9).

**SA-BMA (MCMC)**    For SGLD, it is unnecessary to convert pre-trained DNN into BNN. Instead, we directly set the pre-trained DNN as initialization. We incorporated noise into Eq. (9) without the KLD term ($\beta = 0$) based on the learning rate and the hyperparameter, temperature. During the first step, the adversarial posterior is computed without any noise (Eq. (7)). In the second step, both the noise and the adversarial posterior are used together in the learning process.

**SA-BMA (SWAG)**    SWAG is also one of the options to convert pre-trained DNN into BNN. Specifically, we run a few epochs with source or downstream datasets to make BNN from pre-trained DNN. After this step, we set the BNN as the prior, $q_\theta^{\mathrm{pr}}(w|\mathcal{D}^{\mathrm{pr}})$ in Eq. (9). We also let the converted BNN as initialization and train with downstream dataset. We optimize model with the loss function in Eq. (9).

## D.2    HYPERPARAMETERS FOR EXPERIMENTS

In this section, we provide the details of the experimental setup for Section 5.3. In the other experiments, the range of hyperparameters, excluding the number of epochs, is shared across different backbones and methods.

First, we provide remarks for each baseline method, followed by the tables of hyperparameter configuration with respect to downstream datasets and the baselines. For all experiments, the hyperparameters are selected using grid-search. Configuration of best hyperparameters for each baseline is summarized in Table 12 and Table 13. We ran all experiments using GeForce RTX 3090 and NVIDIA RTX A6000 with GPU memory of 24,576MB and 49,140 MB.

**Stochastic Gradient Descent with Momentum (SGD)**    In this study, we adopt Stochastic Gradient Descent with Momentum as an optimizer for DNN. Learning rate schedule is fixed to cosine decay with warmup length of 10. We tested [100, 150] epoch and set 100 epoch as the best option. In overall experiments, we set momentum as 0.9. The hyperparameter tuning range included learning rate in [1e-4, 1e-3, 1e-2], and weight decay in [1e-4, 5e-4, 1e-3, 1e-2].

**Sharpness Aware Minimization (SAM)**    We set SGD with momentum as the base optimizer of SAM. It also ran upon a cosine decay learning rate scheduler. All the range of hyperparameters is shared with SGD with Momenmtum. Additional hyperparameter $\gamma$, the ball size of perturbation, is in [1e-2, 5e-2, 1e-1].

**Fisher SAM (FSAM)**    We set SGD with momentum as the base optimizer of FSAM. It also ran upon a cosine decay learning rate scheduler. All the range of hyperparameters is shared with SGD with Momenmtum. Additional hyperparameter $\eta$, regularize Fisher impact, is in [1e-2, 1e-1, 1].

**SAM as an optimal relaxation of Bayes (bSAM)**    We use a cosine learning rate decay scheme, annealing the learning rate to zero. We fine-tuned pre-trained models for 150 epochs with fixed $\beta_1$

Table 12: Hyperparameter Configuration for CIFAR10

| Backbone | Baseline | learning rate | $\beta_1$ (momentum) | $\beta_2$ | $\gamma$ | weight decay |
|---|---|---|---|---|---|---|
| RN18 | SGD | 5e-3 | 9e-1 | $\times$ | $\times$ | 1e-3 |
| | SAM | 1e-2 | 9e-1 | $\times$ | 1e-1 | 1e-4 |
| | FSAM | 1e-2 | 9e-1 | $\times$ | 1e-1 | 1e-4 |
| | bSAM | 1e-1 | 9e-1 | 0.999 | 5e-2 | 1e-1 |
| | MOPED | 1e-2 | 9e-1 | $\times$ | $\times$ | 1e-4 |
| | **SA-BMA (VI)** | 1e-2 | 9e-1 | $\times$ | 7e-1 | 1e-3 |
| | MCMC | 5e-2 | 9e-1 | $\times$ | $\times$ | 5e-4 |
| | PTL | 1e-1 | $\times$ | $\times$ | $\times$ | 1e-3 |
| | E-MCMC | 5e-2 | $\times$ | $\times$ | $\times$ | 1e-3 |
| | **SA-BMA (MCMC)** | 5e-3 | 9e-1 | $\times$ | 8e-3 | 5e-4 |
| | SWAG | 5e-3 | 9e-1 | $\times$ | $\times$ | 1e-5 |
| | F-SWAG | 5e-3 | 9e-1 | $\times$ | 5e-2 | 5e-4 |
| | **SA-BMA (SWAG)** | 5e-2 | 9e-1 | $\times$ | 1e-1 | 5e-4 |
| ViT-B/16 | SGD | 1e-3 | 9e-1 | $\times$ | $\times$ | 1e-4 |
| | SAM | 1e-3 | 9e-1 | $\times$ | 1e-2 | 1e-3 |
| | FSAM | 5e-3 | 9e-1 | $\times$ | 1e-2 | 1e-3 |
| | bSAM | 1e-1 | 9e-1 | 0.999 | 1e-2 | 1e-1 |
| | MOPED | 1e-3 | 9e-1 | $\times$ | $\times$ | 1e-4 |
| | **SA-BMA (VI)** | 1e-2 | 9e-1 | $\times$ | 1e-1 | 5e-4 |
| | MCMC | 3e-2 | 9e-1 | $\times$ | $\times$ | 5e-4 |
| | PTL | 6e-2 | $\times$ | $\times$ | $\times$ | 1e-3 |
| | EMCMC | 5e-3 | $\times$ | $\times$ | $\times$ | 1e-2 |
| | **SA-BMA (MCMC)** | 5e-3 | 9e-1 | $\times$ | 8e-3 | 5e-4 |
| | SWAG | 1e-3 | 9e-1 | $\times$ | $\times$ | 1e-3 |
| | F-SWAG | 1e-3 | 9e-1 | $\times$ | 1e-2 | 1e-3 |
| | **SA-BMA (SWAG)** | 5e-3 | 9e-1 | $\times$ | 5e-1 | 5e-4 |

and $\beta_2$. The hyperparameter tuning range included: learning rate in [1e-3, 1e-2, 5e-2, 1e-1, 0.25, 0.5, 1], weight decay in [1e-3, 1e-2, 1e-1], damping in [1e-3, 1e-2, 1e-1], noise scaling parameter in [1e-4, 1e-3, 1e-2, 1e-1], and $\gamma$ in [1e-3, 1e-2, 5e-2, 1e-1]. Damping parameter stabilizes the method by adding constant when updating variance estimate. Since SAM as Bayes optimizer depends on the number of samples to scale the prior, we introduced additional noise scaling parameters to mitigate the gap between the experimental settings, where SAM as Bayes assumed training from scratch and our method assumed few-shot fine-tuning on the pre-trained model. We multiplied noise scaling parameter to the variance of the Gaussian noise to give strong prior, assuming pre-trained model.

**Model Priors with Empirical Bayes using DNN (MOPED)**  MOPED was a baseline to compare for Bayesian Transfer Learning. It employs pre-trained DNN and transforms it into Mean-Field Variational Inference (MFVI). We set prior mean and variance as 0 and 1, respectively. Besides, we set the posterior mean as 0 and variance as 1e-3. We adopt Reparameterization as type of VI. The essential hyperparameter for MOPED is $\delta$, which adjusts how much to incorporate pre-trained weights. The $\delta$ was searched in [5e-2, 1e-1, 2e-1]. Moreover, we add a hyperparameter $\beta$ for MOPED that can balance the loss term in VI. The $\beta$ is in range [1e-2, 1e-1, 1].

**MCMC**  We consistently use SGLD (Welling & Teh, 2011) for MCMC in this work. It ran upon a cyclic cosine decay learning rate scheduler. The number of cycles was ranged in [2, 4]. The number of sampled models is in [10, 20, 28]. We search temperature in [1e-5, 1e-4, 1e-3, 1e-2, 1e-1, 1].

**Pre-train Your Loss (PTL)**  The backbones both ResNet18 and Vit-B/16 were refined through fine-tuning with a classification head for the target task, leveraging a prior distribution learned from SWAG on the ImageNet 1k dataset using SGD. First, the hyperparameter tuning range of the pre-training epoch is [2, 3, 5, 15, 30] to generate the prior distribution on the source task, ImageNet 1k. The learning rate was 0.1. We approximated the covariance low rank as 5. Second, in the downstream

Table 13: Hyperparameter Configuration for CIFAR100

| Backbone | Baseline | learning rate | $\beta_1$ (momentum) | $\beta_2$ | $\gamma$ | weight decay |
|---|---|---|---|---|---|---|
| | SGD | 1e-2 | 9e-1 | $\times$ | $\times$ | 5e-3 |
| | SAM | 1e-2 | 9e-1 | $\times$ | 5e-2 | 1e-2 |
| | FSAM | 1e-2 | 9e-1 | $\times$ | 1e-1 | 1e-4 |
| | bSAM | 1 | 9e-1 | 0.999 | 1e-2 | 1e-2 |
| RN18 | MOPED | 1e-2 | 9e-1 | $\times$ | $\times$ | 1e-3 |
| | **SA-BMA (VI)** | 5e-2 | 9e-1 | $\times$ | 1e-2 | 5e-4 |
| | MCMC | 3e-2 | 9e-1 | $\times$ | $\times$ | 5e-4 |
| | PTL | 5e-1 | $\times$ | $\times$ | $\times$ | 1e-3 |
| | E-MCMC | 5e-2 | $\times$ | $\times$ | $\times$ | 1e-3 |
| | **SA-BMA (MCMC)** | 1e-2 | 9e-1 | $\times$ | 1e-1 | 5e-4 |
| | SWAG | 1e-2 | 9e-1 | $\times$ | $\times$ | 1e-4 |
| | F-SWAG | 1e-2 | 9e-1 | $\times$ | 5e-2 | 1e-2 |
| | **SA-BMA (SWAG)** | 5e-2 | 9e-1 | $\times$ | 5e-1 | 5e-4 |
| | SGD | 1e-3 | 9e-1 | $\times$ | $\times$ | 1e-2 |
| | SAM | 1e-3 | 9e-1 | $\times$ | 1e-2 | 1e-2 |
| | FSAM | 5e-3 | 9e-1 | $\times$ | 1e-2 | 1e-4 |
| | bSAM | 2.5e-1 | 9e-1 | 0.999 | 1e-2 | 1e-3 |
| ViT-B/16 | MOPED | 1e-3 | 9e-1 | $\times$ | $\times$ | 1e-3 |
| | **SA-BMA (VI)** | 1e-2 | 9e-1 | $\times$ | 5e-2 | 5e-4 |
| | MCMC | 5e-2 | 9e-1 | $\times$ | $\times$ | 5e-4 |
| | PTL | 1e-1 | $\times$ | $\times$ | $\times$ | 1e-3 |
| | E-MCMC | 5e-2 | $\times$ | $\times$ | $\times$ | 1e-3 |
| | **SA-BMA (MCMC)** | 8e-3 | 9e-1 | $\times$ | 8e-3 | 5e-4 |
| | SWAG | 1e-3 | 9e-1 | $\times$ | $\times$ | 1e-2 |
| | F-SWAG | 1e-3 | 9e-1 | $\times$ | 1e-2 | 1e-2 |
| | **SA-BMA (SWAG)** | 1e-2 | 9e-1 | $\times$ | 5e-1 | 5e-4 |

task, the fine-tuning optimizer is SGLD with a cosine learning rate schedule, sampling 30 in 5 cycles. The hyperparameter tuning range included: learning rate in [1e-4, 1e-3, 1e-2, 5e-2, 6e-2, 1e-1, 5e-1], weight decay in [1e-4, 1e-3 ,1e-2 ,1e-1], and prior scale in [1e+4, 1e+5, 1e+6]. Prior scaling in the downstream task is to reflect the mismatch between the pre-training and downstream tasks and to add coverage to parameter settings that might be consistent with the downstream. Training was conducted over 150 epochs; tuning range of fine-tuning epoch is [100, 150, 200, 300, 1000].

**Entropy-MCMC (E-MCMC)** We use a cosine learning rate decay scheme, annealing the learning rate to zero. We set the range of the hyperparameter sweep to the surroundings of the best hyperparameter in E-MCMC for ResNet18: learning rate in [5e-3, 5e-2, 5e-1], weight decay in [1e-4, 1e-3, 1e-2], $\eta$ in [1e-6, 5e-6, 1e-5, 5e-5, 1e-4, 4e-4, 5e-3, 8e-3, 1e-2] and a system temperature $T$ in [1e-5, 1e-4, 1e-3]. In this study, we performed an extensive exploration of the hyperparameter space of ViT-B/16, as it has a mechanism different from the CNN family and may not be found near the best hyperparameter range of ResNet18: learning rate in [1e-3, 5e-3, 1e-2, 5e-2, 5e-1], weight decay in [1e-5, 1e-4, 5e-4, 1e-3, 1e-2, 5e-2], $\eta$ in [5e-7, 1e-6, 5e-6, 5e-5, 1e-4, 4e-4, 5e-4, 1e-3, 8e-3, 1e-2, 1e-1] and a system temperature $T$ in [1e-6, 5e-6, 1e-5, 5e-5, 1e-4, 1e-3, 1e-2, 1e-1]. We fine-tuned pre-trained models for 150 epochs. Note that the $\eta$ handles flatness, and the system temperature adjusts the weight update's step size.

**SWAG** We use a cosine learning rate decay scheme for SWAG. All the range of hyperparameters is shared with SGD with Momenmtum. Additionally, we search three additional hyperparameters for SWAG, capturing DNN snapshots and calculating statistics. First, the epoch to start SWA is in [51, 76, 101] and epoch is in [100, 150]. Second, the frequency to capture the model snapshot is in [1, 2, 3]. Third, the low rank for covariance is in [2, 3, 5, 7, 10].

**F-SWAG** F-SWAG shares hyperparameter with SWAG, except $\gamma$. We search $\gamma$ in [1e-2, 5e-2, 1e-1].

**Sharpness-aware Bayesian Model Averaging (SA-BMA)**  In case of SA-BMA (SWAG), we train SWAG on source task IN 1K to make prior distribution and follow the pre-training protocol of PTL. In case of employing MOPED to make prior distribution, we do not go through any training step. In case of SA-BMA (VI), we just set $\delta$ as 0.05 for MOPED and make DNN into BNN. In case of SA-BMA (MCMC), we just set pre-trained weight as initialization and run experiments. After getting prior distribution, we search three hyperparameters: learning rate, $\gamma$, and $\alpha$. The hyperparamter tuning range included: learning rate in [1e-3, 5e-3, 1e-2, 5e-2], $\gamma$ in [5e-3, 8e-3, 1e-2, 5e-2, 1e-1, 5e-1, 7e-1], and $\alpha$ in [1e-6, 1e-5, 1e-4, 1e-3]. We set weight decay as $5e-4$ for all backbones and train the model over 150 epochs with early stopping. We fix $\beta$ as 1e-8 for all experiments.

## D.3   ALGORITHM OF SA-BMA

Training algorithm of SA-BMA with Bayesian transfer learning can be depicted as Algorithm 1. In the first step, load a model pre-trained on the source task. Note that the pre-trained models do not have to be BNN. Namely, it is capable of using DNN, which can be easier to find than pre-trained BNN. Second, change the loaded DNN into BNN on the source or downstream task. Every BNN framework, containing VI, SWAG, LA, etc., can be adopted to make DNN into BNN. This study mainly employs PTL (Shwartz-Ziv et al., 2022) and MOPED (Krishnan et al., 2020) for this step. We can skip this second step if you load a pre-trained BNN model before. Third, train the subnetwork of the converted BNN model with the proposed flat-seeking seeking optimizer. It allows model to converge into flat minina efficiently.

---

**Algorithm 1** SA-BMA with Bayesian Transfer Learning

---

**Require:** Variational parameter $\theta$, Neighborhood size $\gamma$, Epochs $E$, and Learning rate $\eta_{\text{SA-BMA}}$

  1) Load pre-trained DNN
  2) Make pre-trained DNN model into BNN $q_\theta^{\text{pr}}(w|\mathcal{D}^{\text{pr}})$ and set as prior
 **for** $t = 1, 2, ..., E$ **do**
   3-1) $w \sim q_\theta(w|\mathcal{D}^{\text{ft}})$             $\triangleright$ Sample weight from posterior
   3-4) Forward and calculate the loss $l(\theta)$ with the sampled $w$
   3-5) Backward pass and compute $\nabla_\theta \log p_\theta(w|\mathcal{D})$
   3-6) Compute $F_\theta^{-1}(\theta) = \frac{\nabla_\theta \log p_\theta(w|\mathcal{D}) \nabla_\theta \log p_\theta(w|\mathcal{D})^T}{\|\nabla_\theta \log p_\theta(w|\mathcal{D})\|^4}$
   3-7) Compute the perturbation $\Delta\theta_{\text{SA-BMA}} = \gamma \frac{F_\theta(\theta)^{-1} \nabla_\theta l(\theta)}{\sqrt{\nabla_\theta l(\theta)^T F_\theta(\theta)^{-1} \nabla_\theta l(\theta)}}$
   3-8) Compute gradient approximation for the SA-BMA $\nabla_\theta l_{\text{SA-BMA}}(\theta) = \frac{\partial l(\theta)}{\partial \theta}|_{\theta + \Delta\theta_{\text{SA-BMA}}}$

   3-9) Update $\theta \rightarrow \theta - \eta \nabla_\theta l_{\text{SA-BMA}}(\theta)$
 **end for**

---

## D.4   EFFICIENCY OF SA-BMA WITH BAYESIAN TRANSFER LEARNING

BNN often struggles with high computation and memory complexity, which makes optimizing large-scale BNN hard. However, SA-BMA only optimizes the last (classifier) and normalization layer, which only requires vector-sized learnable parameters. Table 14 provides the scalability of SA-BMA and baselines in the fine-tuning stage given pre-trained model. SA-BMA only requires fewer learnable parameters since $p_1 \ll p$ and low rank $K$ are even fewer than DNN, where $p_1$ denotes the number of parameters in normalization and last layers. It only needs 1% of learnable parameters compared to other methods in case of RN18 and ViT-B/16. SA-BMA efficiently adapts the model in a few-shot setting.

Table 14: Efficiency of SA-BMA with Bayesian Transfer Learning.

| Method | Optim | Num. of Tr Param. |
|--------|-------|-------------------|
| DNN | SGD | $p$ |
|  | SAM | $p$ |
|  | FSAM | $p$ |
| SWAG | SGD | $p$ |
| F-SWAG | SAM | $p$ |
| VI | bSAM | $p$ |
| MOPED | SGD | $2p$ |
| E-MCMC | SGLD | $2p$ |
| PTL | SGLD | $p$ |
| SA-BMA | SA-BMA | $(K+2)p_1$ |

# E   PROOF AND DERIVATION

## E.1   PROOF OF THEOREM 1

The derivation of Theorem 1 can be straightforward using Wely's inequality (Weyl, 1912).

We assume $M$ model $w_m, m = 1, .., M$, whose Hessian matrices $H_{w_m}$ are Hermitian. $w_{\text{avg}} = 1/M \sum_{m=1}^{M} w_m$ is simple weight averaging and the Hessian of $w_{\text{avg}}$ also be a Hermitian matrix. Let's say $\lambda_n(H_{w_m})$ is $n$-th maximal eigenvalue of $H_{w_m}$ and assume there are $N$ eigenvalues. $\lambda_{\max}(H_{w_m})$ is same as $\lambda_1(H_{w_m})$. Weyl's inequality (Theorem 3) is known to bound the eigenvalues of Hermitian matrices.

**Theorem 3.** *(Weyl's Inequality) For Hermitian matrices* $C_m \in \mathbb{C}^{p \times p}$, $k, l = 1, ..., M$,

$$\lambda_{k+l-1}(C_i + C_j) \leq \lambda_k(C_i) + \lambda_l(C_j) \leq \lambda_{k+l-N}(C_i + C_j). \tag{10}$$

Let $k = 1$ and $l = 1$, then Eq. (10) can be written as:

$$\lambda_1(C_i + C_j) \leq \lambda_1(C_i) + \lambda_1(C_j).$$

As we have $M$ Hermitian matrices, it can be expanded as:

$$\lambda_1\left(\frac{1}{M}\sum_{m=1}^{M} H_{w_m}\right) \leq \frac{1}{M}\sum_{m=1}^{M} \lambda_1(H_{w_m}). \tag{11}$$

One the other hand, we can let $(k, l) = \{(1, N), (N, 1)\}$ and rewrite the Eq. (10) as:

$$\max\{\lambda_1(C_i) + \lambda_N(C_j), \lambda_N(C_i) + \lambda_1(C_j)\} \leq \lambda_1(C_i + C_j).$$

Again, set $M$ Hermitian matrices we have, it can be expanded as:

$$\max\left(\left\{\frac{1}{M}\left(\lambda_1(H_{w_m}) + \sum_{\substack{n=1 \\ n \neq m}}^{M} \lambda_N(H_{w_n})\right)\right\}_{m=1}^{M}\right) \leq \lambda_1\left(\frac{1}{M}\sum_{m=1}^{M} H_{w_m}\right). \tag{12}$$

By combining Eq. (11) with Eq. (12) and substituting $\lambda_1$ to $\lambda_1 \max$ and $\lambda_N$ to $\lambda_{\min}$, the flatness of averaged weight parameter is bounded as:

$$\max\left(\left\{\frac{1}{M}\left(\lambda_{\max}(H_{w_m}) + \sum_{\substack{n=1 \\ n \neq m}}^{M} \lambda_{\min}(H_{w_n})\right)\right\}_{m=1}^{M}\right) \leq \lambda_{\max}\left(\frac{1}{M}\sum_{m=1}^{M} H_{w_m}\right) \leq \frac{\sum_{m=1}^{M} \lambda_{\max}(H_{w_m})}{M}. \tag{13}$$

BMA marginalizes diverse predictions by ensembling model output. As shown in Lemma 1, it is closely related to weight averaging (WA) (Izmailov et al., 2018; Wortsman et al., 2022; Rame et al., 2022).

**Lemma 1.** *((Rame et al., 2022)) Given predictions of model* $f_m(\cdot)$ *parameterized by* $\{w_m\}_{m=1}^{M}$, $w_{WA} = \frac{1}{M}\sum_{m=1}^{M} w_m$, *prediction of averaged model* $f_{WA}$ *parameterized by* $w_{WA}$, *prediction of BMA* $f_{BMA}$, *and arbitrary twice differentiable loss function* $l(\cdot)$, *let* $\Delta = \|f_{BMA}(x) - f_{WA}(x)\|_2$. *Then,* $\forall(x, y)$

$$l(f_{WA}(x), y) = l(f_{BMA}(x), y) + O(\Delta).$$

As the predictions of BMA and WA get closer, we can say the Hessian of loss for BMA and WA become approximately identical. Specifically, they becomes equivalent as $O(\Delta)$ goes to zero, where the predictions of BMA and WA are same.

### E.2 DERIVATION OF BAYESIAN FLAT-SEEKING OPTIMIZER

#### E.2.1 SETTING

Let model parameter $w \subseteq \mathbb{R}^p$ and $w \sim \mathcal{N}(\mu, \Sigma)$. While fully-factorized or mean-field covariance is de facto in Bayesian Deep Learning, it cannot capitalize on strong points of Bayesian approach. Inspired from SWAG, we approximate covariance combining diagonal covariance $\sigma \subseteq \mathbb{R}^p$ and low-rank matrix $L \subseteq \mathbb{R}^{p \times K}$ with low-rank component $K$. Then, we can simply sample $w = \mu + \frac{1}{\sqrt{2}}(\sigma z_1 + L z_2)$, where $z_1 \sim \mathcal{N}(0, I_p)$ and $z_2 \sim \mathcal{N}(0, I_K)$ where $p$, $K$ denotes the number of parameter, low-rank component, respectively. We treat flattened $\mu$, $\sigma$, and $L$, and concatenate as $\theta = \text{Concat}(\mu; \sigma; L)$.

#### E.2.2 OBJECTIVE FUNCTION

We compose our objective function with probabilistic weight, using KL Divergence as a metric to compare between two weights.

$$l_{\text{SA-BMA}}^{\gamma}(\theta) = \max_{d|\theta + \Delta\theta, \theta| \leq \gamma^2} l(\theta + \Delta\theta) + \beta D_{\text{KL}}(p_\theta(w|\mathcal{D})||p(w)) \tag{14}$$

$$\text{s.t. } d|\theta + \Delta\theta, \theta| = D_{\text{KL}}\big[p_{\theta+\Delta\theta}(w|\mathcal{D})||p_\theta(w|\mathcal{D})\big]. \tag{15}$$

#### E.2.3 OPTIMIZATION

**From KL Divergence to Fisher Information Matrix**   We can consider three options of perturbation on mean and covariance parameters of $w$: 1) Perturbation on mean, 2) perturbation on mean and diagonal variance, 3) Perturbation on mean and whole covariance. All of them can be approximated to Fisher Information Matrix. Here, we show the relation between KLD and FIM considering the probation option 3.

Following FSAM, we deal with parameterized and conditioned as same notation:

$$p_{\theta+\Delta\theta}(w|\mathcal{D}) = p(w|\mathcal{D}, \theta + \Delta\theta).$$

By definition of KL divergence, we rewrite Eq. (15) as:

$$D_{\text{KL}}[p(w|\mathcal{D}, \theta + \Delta\theta)||p(w|\mathcal{D}, \theta)] = \int_w p(w|\mathcal{D}, \theta + \Delta\theta) \, \log \frac{p(w|\mathcal{D}, \theta + \Delta\theta)}{p(w|\mathcal{D}, \theta)} dw. \tag{16}$$

In Eq. (16), we apply first-order Taylor Expansion:

$$p(w|\mathcal{D}, \theta + \Delta\theta) \approx p(w|\mathcal{D}, \theta) + \nabla_\theta p(w|\mathcal{D}, \theta)^T \Delta\theta.$$
$$\log p(w|\mathcal{D}, \theta + \Delta\theta) \approx \log p(w|\mathcal{D}, \theta) + \nabla_\theta \log p(w|\mathcal{D}, \theta)^T \Delta\theta. \tag{17}$$

Substitute right terms of Eq. (16) with Eq. (17):

$$\int_w p(w|\mathcal{D}, \theta + \Delta\theta) \, \log \frac{p(w|\mathcal{D}, \theta + \Delta\theta)}{p(w|\mathcal{D}, \theta)} dw$$
$$= \int_w \big(p(w|\mathcal{D}, \theta) + \Delta\theta^T \nabla_\theta p(w|\mathcal{D}, \theta)\big) \nabla_\theta \log p(w|\mathcal{D}, \theta)^T \Delta\theta \, dw$$
$$= \int_w p(w|\mathcal{D}, \theta) \nabla_\theta \log p(w|\mathcal{D}, \theta)^T \Delta\theta dw$$
$$+ \int_w \Delta\theta^T p(w|\mathcal{D}, \theta) \nabla_\theta \log p(w|\mathcal{D}, \theta) \nabla_\theta \log p(w|\mathcal{D}, \theta)^T \Delta\theta \, dw. \tag{18}$$

First term of Eq. (18) is equal to 0:

$$\int_w p(w|\mathcal{D}, \theta) \nabla_\theta \log p(w|\mathcal{D}, \theta) \, dw$$

$$= \int_w p(w|\mathcal{D}, \theta) \frac{\nabla_\theta p(w|\mathcal{D}, \theta)}{p(w|\mathcal{D}, \theta)} \, dw \tag{19}$$

$$= \int_w \nabla_\theta p(w|\mathcal{D}, \theta) \, dw \ = \nabla_\theta \int_w p(w|\mathcal{D}, \theta) = 0.$$

We can rewrite Eq. (16) using Eq. (18), Eq. (19) and find it's related to Fisher information matrix by the definition of expectation:

$$D_{KL}[p(w|\mathcal{D}, \theta + \Delta\theta)||p(w|\mathcal{D}, \theta)]$$

$$= \int_w \Delta\theta^T p(w|\mathcal{D}, \theta) \nabla_\theta \log p(w|\mathcal{D}, \theta) \nabla_\theta \log p(w|\mathcal{D}, \theta)^T \Delta\theta \tag{20}$$

$$= \Delta\theta^T \mathbb{E}_w[\nabla_\theta \log p(w|\mathcal{D}, \theta) \nabla_\theta \log p(w|\mathcal{D}, \theta)^T] \Delta\theta$$

$$= \Delta\theta^T F_\theta(\theta) \Delta\theta,$$

where $F_\theta(\theta) = \mathbb{E}_{w, \mathcal{D}}[\nabla_\theta \log p(w|\mathcal{D}, \theta) \nabla_\theta \log p(w|\mathcal{D}, \theta)^T]$.

It's too expensive to calculate Fisher information matrix $F(\theta)$ in practice. We introduce a pseudo inverse for Fisher information matrix $F_\theta(\theta)^{-1}$ with Samelson inverse of a vector (Gentle, 2007; Sidi, 2017; Wynn, 1962) :

$$F_\theta(\theta)^{-1} = \frac{\nabla_\theta \log p(w|\mathcal{D}, \theta) \nabla_\theta \log p(w|\mathcal{D}, \theta)^T}{\|\nabla_\theta \log p(w|\mathcal{D}, \theta)\|^4}. \tag{21}$$

**Lagrangian Dual Problem**   From the result of Eq. (20), we can rewrite the Eq. (14):

$$l_{\text{SA-BMA}}^\gamma(\theta) = \max_{\Delta\theta^T F_\theta(\theta) \Delta\theta \leq \gamma^2} l(\theta + \Delta\theta). \tag{22}$$

We can reach the optimal perturbation of SA-BMA $\Delta\theta^*$ by using Taylor Expansion on $l(\theta + \Delta\theta)$ of Eq. (14):

$$l(\theta + \Delta\theta) = l(\theta) + \nabla_\theta l(\theta)^T \Delta\theta. \tag{23}$$

Using Eq. (23), we can rewrite Eq. (14) as Lagrangian dual problem:

$$L(\Delta\theta, \lambda) = l(\theta) + \nabla l_\theta(\theta)^T \Delta\theta - \lambda(\Delta\theta^T F_\theta(\theta) \Delta\theta - \gamma^2). \tag{24}$$

Differentiating Eq. (24), we get $\Delta\theta^*$:

$$\frac{\alpha L(\Delta\theta, \lambda)}{\alpha \Delta\theta} = \nabla_\theta l(\theta)^T - 2\lambda \Delta\theta^T F_\theta(\theta) = 0$$

$$\therefore \ \Delta\theta^* = \frac{1}{2\lambda} F_\theta(\theta)^{-1} \nabla_\theta l(\theta). \tag{25}$$

Putting $\Delta\theta^*$ of Eq. (25) into $\Delta\theta$ of Eq. (24), we can rewrite Eq. (24):

$$L(\Delta\theta^*, \lambda) = l(\theta) + \frac{1}{2\lambda} \nabla_\theta l(\theta)^T F_\theta(\theta)^{-1} \nabla_\theta l(\theta)$$

$$- \frac{1}{4\lambda} \nabla_\theta l(\theta)^T F_\theta(\theta)^{-1} \nabla_\theta l(\theta) + \lambda\gamma^2. \tag{26}$$

By taking derivative of Eq. (26) w.r.t. $\lambda$, we can also get $\lambda^*$:

$$
\begin{aligned}
\frac{\alpha L(\Delta\theta^*, \lambda)}{\alpha\lambda} &= -\frac{1}{2\lambda^2}\nabla_\theta l(\theta)^T F_\theta(\theta)^{-1}\nabla_\theta l(\theta) + \frac{1}{4\lambda^2}\nabla_\theta l(\theta)^T F_\theta(\theta)^{-1}\nabla_\theta l(\theta) + \gamma^2 = 0 \\
4\lambda^2\gamma^2 &= \nabla_\theta l(\theta)^T F_\theta(\theta)^{-1}\nabla_\theta l(\theta) \\
\therefore\ \lambda^* &= \frac{\sqrt{\nabla_\theta l(\theta)^T F_\theta(\theta)^{-1}\nabla_\theta l(\theta)}}{2\gamma}.
\end{aligned}
\tag{27}
$$

Finally, we get our $\Delta\theta^*_{SA-BMA}$ by substituting Eq. (27) into Eq. (25):

$$
\Delta\theta^*_{\text{SA-BMA}} = \gamma\frac{F_\theta(\theta)^{-1}\nabla_\theta l(\theta)}{\sqrt{\nabla_\theta l(\theta)^T F_\theta(\theta)^{-1}\nabla_\theta l(\theta)}}.
\tag{28}
$$

### E.3 PROOF OF THEOREM 2

#### E.3.1 SA-BMA TO FSAM

Theorem 2 shows that SA-BMA is degenerated to FSAM under DNN and diagonal FIM setting. Deterministic parameters draw out the constant prior $p(w|x) = c$ and mean-only variational parameters $w = \theta$.

First, we can rewrite the log posterior $\log p_\theta(w|x, y)$ with Bayes rule:

$$
\log p_\theta(w|x, y) = \log p_\theta(y|x, w) + \log p_\theta(w|x) - Z,
\tag{29}
$$

where $Z$ is constant independent of $w$. Is is noted that the log posterior is divided into the log predictive distribution and log prior. Also, note that the prior is conditioned on the data to align with a generalized notation. The prior can depend on the input; however, this dependence is often ignored in practice (Marek et al., 2024).

By taking derivative with respect to $\theta$ on Eq. (29), the constant $Z$ goes to 0:

$$
\nabla_\theta \log p_\theta(w|x, y) = \nabla_\theta p_\theta(y|x, w) + \nabla_\theta \log p_\theta(w|x).
$$

We have constant prior $p(w|x) = c$ in deterministic setting and it makes the gradient of log posterior and log predictive distribution:

$$
\nabla_\theta \log p_\theta(w|x, y) = \nabla_\theta p_\theta(y|x, w).
\tag{30}
$$

Underlying Eq. (30), it is possible to substitute the gradient of log posterior into the gradient of log predictive distribution and FIM over posterior goes to FIM over predictive distribution:

$$
\begin{aligned}
F_\theta(\theta) &= \mathbb{E}_{w,\mathcal{D}}[\nabla_\theta \log p_\theta(w|x, y)\nabla_\theta \log p_\theta(w|x, y)^T] \\
&= \mathbb{E}_{w,\mathcal{D}}[\nabla_\theta \log p_\theta(y|x, w)\nabla_\theta \log p_\theta(y|x, w)^T].
\end{aligned}
\tag{31}
$$

By taking diagonal computation over Eq. (31), it goes to $F_y(\theta)$. After that, using the fact that mean-only variational parameters, SA-BMA degnerates to FSAM with $F_y(\theta)$ finally.

$$
\Delta\theta_{\text{SA-BMA}} = \gamma\frac{F_y(\theta)^{-1}\nabla_\theta l(\theta)}{\sqrt{F_y(\theta)^{-1}\nabla_\theta l(\theta)F_y(\theta)^{-1}}}.
\tag{32}
$$

#### E.3.2 SA-BMA TO SAM

It is simple to show that SA-BMA is extended version of SAM by defining FIM over output distribution $F_y(w)$ as identity matrix $I$ in Eq. (32), SA-BMA goes to SAM.

$$
\Delta\theta_{\text{SA-BMA}} = \gamma\frac{\nabla_w l(w)}{\|\nabla_w l(w)\|_2}.
\tag{33}
$$

### E.3.3 SA-BMA TO NG

Theorem 2 also states the NG can be approximated with SA-BMA under specific conditions. The update rule of natural gradient and SA-BMA can be written as Eq. (34) and Eq. (35), respectively.

$$\theta \leftarrow \theta + \eta_{\text{NG}} F_y(\theta)^{-1} \nabla_\theta l(\theta). \tag{34}$$

$$\theta \leftarrow \theta + \eta_{\text{SA-BMA}} \nabla_\theta l(\theta + \Delta\theta). \tag{35}$$

where $\eta_{\text{NG}}$ and $\eta_{\text{SA-BMA}}$ denote the learning rate of NG and SA-BMA. Note that we assume the log likelihood as loss fuction.

The $\nabla_\theta l(\theta + \Delta\theta)$ in Eq. (35) can be approximated with Taylor Expansion, the connection between Hessian and FIM, and Eq. (31) in DNN setup:

$$
\begin{aligned}
\nabla_\theta l(\theta + \Delta\theta) &\approx \nabla_\theta l(\theta) + \nabla_\theta^2 \Delta\theta \\
&= \nabla_\theta l(\theta) + \nabla_\theta^2 l(\theta) \cdot \gamma \frac{F_\theta(\theta)^{-1} \nabla_\theta l(\theta)}{\sqrt{\nabla_\theta l(\theta)^T F_\theta(\theta)^{-1} \nabla_\theta l(\theta)}} \\
&= \nabla_\theta l(\theta) + \gamma' \nabla_\theta^2 l(\theta) F_\theta(\theta)^{-1} \nabla_\theta l(\theta) \left( \because \text{Let } \gamma' = \frac{\gamma}{\sqrt{\nabla_\theta l(\theta)^T F_\theta(\theta)^{-1} \nabla_\theta l(\theta)}} \right) \\
&= [I + \gamma' \nabla_\theta^2 l(\theta) F_\theta(\theta)^{-1}] \nabla_\theta l(\theta) \\
&\approx (1 + \gamma') \nabla_\theta l(\theta) \ (\because \nabla_\theta^2 l(\theta) \approx F_y(\theta), F_\theta(\theta) = F_y(\theta)).
\end{aligned}
\tag{36}
$$

By using the denoted learning rate $\eta_{\text{SA-BMA}} = \frac{\eta_{\text{NG}}}{I + \gamma'} F_\theta(\theta)^{-1}$, Eq. (31), and Eq. (36), update rule of SA-BMA approximates to NG.

## F  FINE-GRAINED IMAGE CLASSIFICATION

In addition to classification accuracy, SA-BMA shows superior performance compared to the baseline in NLL metric, indicating that SA-BMA effectively quantifies uncertainty.

Table 15: Downstream task NLL with RN50 and ViT-B/16 pre-trained on IN 1K. SA-BMA (SWAG) denotes using SWAG to convert pre-trained model into BNN. **Bold** and underline denote best and second best performance each. SA-BMA demonstrates superior performance across all 16-shot datasets, including EuroSAT , Oxford Flowers, Oxford Pets, and UCF101.

| Backbone | | RN50 | | | | | ViT-B/16 | | | | |
|---|---|---|---|---|---|---|---|---|---|---|---|
| Method | Optim | EuroSAT | Oxford Flowers | Oxford Pets | UCF101 | Avg | EuroSAT | Oxford Flowers | Oxford Pets | UCF101 | Avg |
| DNN | SGD | $0.416_{\pm0.043}$ | $0.265_{\pm0.010}$ | $0.367_{\pm0.008}$ | $1.331_{\pm0.024}$ | $0.595_{\pm0.010}$ | $0.573_{\pm0.044}$ | $0.361_{\pm0.027}$ | $0.385_{\pm0.044}$ | $1.246_{\pm0.044}$ | $0.641_{\pm0.020}$ |
| DNN | SAM | $0.376_{\pm0.003}$ | $\underline{0.190}_{\pm0.001}$ | $\underline{0.344}_{\pm0.014}$ | $\underline{1.157}_{\pm0.035}$ | $0.517_{\pm0.005}$ | $0.522_{\pm0.023}$ | $0.276_{\pm0.029}$ | $\underline{0.287}_{\pm0.022}$ | $\underline{1.140}_{\pm0.034}$ | $\underline{0.556}_{\pm0.020}$ |
| SWAG | SGD | $0.343_{\pm0.046}$ | $0.264_{\pm0.011}$ | $0.367_{\pm0.007}$ | $1.347_{\pm0.022}$ | $0.580_{\pm0.009}$ | $0.547_{\pm0.021}$ | $0.361_{\pm0.027}$ | $0.366_{\pm0.010}$ | $1.286_{\pm0.045}$ | $0.640_{\pm0.006}$ |
| F-SWAG | SAM | $\underline{0.301}_{\pm0.039}$ | $0.190_{\pm0.002}$ | $0.351_{\pm0.010}$ | $1.186_{\pm0.034}$ | $\underline{0.507}_{\pm0.008}$ | $0.514_{\pm0.018}$ | $0.276_{\pm0.033}$ | $0.297_{\pm0.030}$ | $1.234_{\pm0.031}$ | $0.580_{\pm0.017}$ |
| MOPED | SGD | $0.481_{\pm0.100}$ | $0.347_{\pm0.019}$ | $0.388_{\pm0.007}$ | $1.367_{\pm0.029}$ | $0.646_{\pm0.028}$ | $\underline{0.484}_{\pm0.018}$ | $0.354_{\pm0.025}$ | $0.309_{\pm0.015}$ | $1.180_{\pm0.028}$ | $0.582_{\pm0.017}$ |
| PTL | SGLD | $0.319_{\pm0.006}$ | $0.307_{\pm0.010}$ | $0.360_{\pm0.015}$ | $1.391_{\pm0.036}$ | $0.594_{\pm0.010}$ | $0.493_{\pm0.012}$ | $0.616_{\pm0.066}$ | $0.381_{\pm0.008}$ | $1.670_{\pm0.050}$ | $0.790_{\pm0.013}$ |
| SABMA (SWAG) | SABMA | $\mathbf{0.297}_{\pm0.038}$ | $\mathbf{0.147}_{\pm0.037}$ | $\mathbf{0.339}_{\pm0.023}$ | $\mathbf{1.113}_{\pm0.009}$ | $\mathbf{0.474}_{\pm0.023}$ | $\mathbf{0.455}_{\pm0.006}$ | $\mathbf{0.219}_{\pm0.037}$ | $\mathbf{0.272}_{\pm0.006}$ | $\mathbf{1.071}_{\pm0.036}$ | $\mathbf{0.504}_{\pm0.012}$ |

## G  PERFORMANCE UNDER DISTRIBUTION SHIFT

We adopt the corrupted dataset CIFAR10/100C to test the robustness over distribution shift. The corrupted dataset transform the CIFAR10/100-test dataset, which has been modified to shift the distribution of the test data further away from the training data. It contains 19 kinds of corrupt options, such as varying brightness or contrast to adding Gaussian noise. The severity level indicates the strength of the transformation and is typically expressed as a number from 1 to 5, where the higher the number, the stronger the transformation. In Figure 13, our method ensures relatively robust performance in the data distribution shift, even as the severity increases.

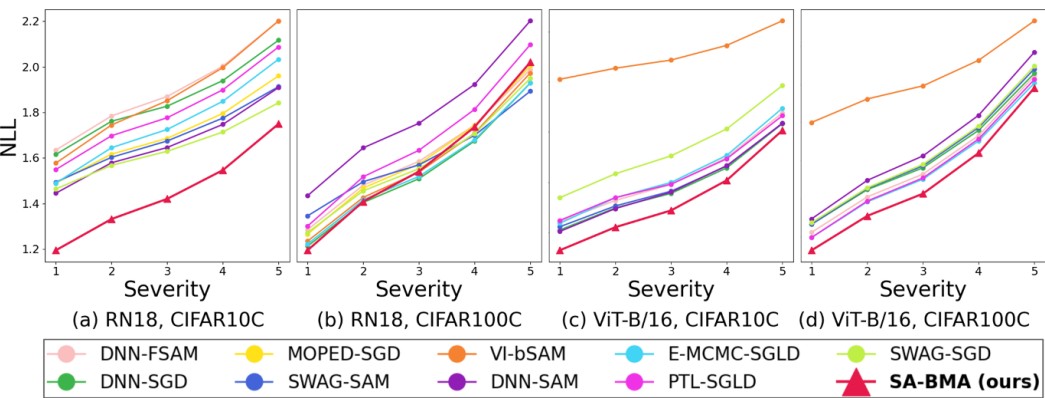

Figure 13: NLL performance of ResNet 18 and ViT-B/16 on corrupted CIFAR10 and CIFAR100, respectively (Hendrycks & Dietterich, 2019).

We also provide the detailed results of three repeated experiments with corrupted sets.

(a) RN18 CIFAR10C

| Method | Optim | Severity | | | | | | | | | |
| | | 1 | | 2 | | 3 | | 4 | | 5 | |
| | | ACC ↑ | NLL ↓ | ACC ↑ | NLL ↓ | ACC ↑ | NLL ↓ | ACC ↑ | NLL ↓ | ACC ↑ | NLL ↓ |
| DNN | SGD | $49.57_{\pm0.97}$ | $1.49_{\pm0.02}$ | $45.78_{\pm1.43}$ | $1.62_{\pm0.04}$ | $43.78_{\pm1.44}$ | $1.69_{\pm0.04}$ | $40.83_{\pm1.59}$ | $1.80_{\pm0.06}$ | $36.30_{\pm1.79}$ | $1.96_{\pm0.08}$ |
| | SAM | $50.23_{\pm2.11}$ | $1.62_{\pm0.07}$ | $46.56_{\pm2.00}$ | $1.76_{\pm0.03}$ | $44.59_{\pm2.26}$ | $1.83_{\pm0.03}$ | $41.85_{\pm2.42}$ | $1.94_{\pm0.04}$ | $37.33_{\pm2.52}$ | $2.12_{\pm0.07}$ |
| | FSAM | $48.76_{\pm4.00}$ | $1.63_{\pm0.03}$ | $45.11_{\pm3.91}$ | $1.78_{\pm0.01}$ | $42.94_{\pm3.88}$ | $1.87_{\pm0.03}$ | $40.06_{\pm3.85}$ | $2.00_{\pm0.08}$ | $35.70_{\pm3.50}$ | $2.20_{\pm0.12}$ |
| SWAG | SGD | $50.05_{\pm0.76}$ | $1.55_{\pm0.09}$ | $46.31_{\pm1.16}$ | $1.70_{\pm0.11}$ | $44.17_{\pm1.07}$ | $1.78_{\pm0.11}$ | $41.20_{\pm1.13}$ | $1.90_{\pm0.13}$ | $36.64_{\pm1.26}$ | $2.09_{\pm0.15}$ |
| F-SWAG | SAM | $51.37_{\pm1.08}$ | $1.49_{\pm0.05}$ | $47.35_{\pm0.71}$ | $1.64_{\pm0.04}$ | $45.16_{\pm0.66}$ | $1.72_{\pm0.06}$ | $42.01_{\pm0.57}$ | $1.85_{\pm0.06}$ | $37.27_{\pm0.64}$ | $2.03_{\pm0.07}$ |
| VI | bSAM | $49.20_{\pm2.40}$ | $1.46_{\pm0.05}$ | $45.35_{\pm1.93}$ | $1.57_{\pm0.04}$ | $43.07_{\pm2.10}$ | $1.63_{\pm0.04}$ | $40.12_{\pm1.74}$ | $1.71_{\pm0.03}$ | $35.50_{\pm1.36}$ | $1.84_{\pm0.02}$ |
| MOPED | SGD | $50.72_{\pm0.80}$ | $1.58_{\pm0.11}$ | $46.87_{\pm0.32}$ | $1.74_{\pm0.11}$ | $44.52_{\pm0.39}$ | $1.85_{\pm0.12}$ | $41.38_{\pm0.29}$ | $2.00_{\pm0.12}$ | $36.73_{\pm0.17}$ | $2.20_{\pm0.10}$ |
| E-MCMC | SGLD | $49.86_{\pm1.54}$ | $1.49_{\pm0.03}$ | $46.17_{\pm1.55}$ | $1.60_{\pm0.04}$ | $44.07_{\pm1.72}$ | $1.67_{\pm0.07}$ | $41.05_{\pm1.65}$ | $1.77_{\pm0.10}$ | $36.53_{\pm1.74}$ | $1.91_{\pm0.13}$ |
| PTL | SGLD | $50.44_{\pm1.65}$ | $1.45_{\pm0.06}$ | $46.22_{\pm1.96}$ | $1.58_{\pm0.09}$ | $44.06_{\pm1.67}$ | $1.65_{\pm0.09}$ | $41.02_{\pm1.66}$ | $1.75_{\pm0.11}$ | $36.14_{\pm1.51}$ | $1.91_{\pm0.13}$ |
| SA-BMA (VI) | | $\mathbf{58.53}_{\pm0.75}$ | $\mathbf{1.19}_{\pm0.02}$ | $\mathbf{53.72}_{\pm0.70}$ | $\mathbf{1.33}_{\pm0.00}$ | $\mathbf{50.61}_{\pm0.84}$ | $\mathbf{1.42}_{\pm0.01}$ | $\mathbf{46.76}_{\pm1.15}$ | $\mathbf{1.55}_{\pm0.03}$ | $\mathbf{40.70}_{\pm1.34}$ | $\mathbf{1.75}_{\pm0.05}$ |

(b) RN18 CIFAR100C

| Method | Optim | Severity | | | | | | | | | |
| | | 1 | | 2 | | 3 | | 4 | | 5 | |
| | | ACC ↑ | NLL ↓ | ACC ↑ | NLL ↓ | ACC ↑ | NLL ↓ | ACC ↑ | NLL ↓ | ACC ↑ | NLL ↓ |
| DNN | SGD | $36.01_{\pm0.86}$ | $2.55_{\pm0.02}$ | $31.81_{\pm0.73}$ | $2.79_{\pm0.06}$ | $29.75_{\pm0.57}$ | $2.91_{\pm0.04}$ | $26.73_{\pm0.25}$ | $3.11_{\pm0.02}$ | $22.20_{\pm0.08}$ | $3.40_{\pm0.00}$ |
| | SAM | $37.94_{\pm0.52}$ | $2.46_{\pm0.02}$ | $33.57_{\pm0.50}$ | $\mathbf{2.69}_{\pm0.03}$ | $31.46_{\pm0.67}$ | $\mathbf{2.82}_{\pm0.03}$ | $28.19_{\pm0.75}$ | $3.02_{\pm0.05}$ | $23.32_{\pm0.69}$ | $3.33_{\pm0.06}$ |
| | FSAM | $36.46_{\pm0.44}$ | $2.53_{\pm0.05}$ | $32.24_{\pm0.36}$ | $2.77_{\pm0.04}$ | $30.19_{\pm0.42}$ | $2.90_{\pm0.03}$ | $27.12_{\pm0.37}$ | $3.10_{\pm0.02}$ | $22.48_{\pm0.39}$ | $3.42_{\pm0.01}$ |
| SWAG | SGD | $35.84_{\pm5.17}$ | $2.62_{\pm0.30}$ | $32.43_{\pm4.55}$ | $2.81_{\pm0.27}$ | $30.71_{\pm4.21}$ | $2.89_{\pm0.25}$ | $28.13_{\pm3.81}$ | $3.05_{\pm0.22}$ | $24.24_{\pm2.99}$ | $\mathbf{3.29}_{\pm0.17}$ |
| F-SWAG | SAM | $37.10_{\pm0.60}$ | $2.49_{\pm0.03}$ | $32.84_{\pm0.62}$ | $2.72_{\pm0.03}$ | $30.59_{\pm0.72}$ | $2.86_{\pm0.04}$ | $27.43_{\pm0.91}$ | $3.06_{\pm0.06}$ | $22.74_{\pm0.93}$ | $3.38_{\pm0.08}$ |
| VI | bSAM | $36.20_{\pm0.59}$ | $2.73_{\pm0.03}$ | $32.48_{\pm0.34}$ | $2.99_{\pm0.03}$ | $30.66_{\pm0.33}$ | $3.12_{\pm0.02}$ | $27.94_{\pm0.14}$ | $3.32_{\pm0.05}$ | $23.66_{\pm0.29}$ | $3.66_{\pm0.06}$ |
| MOPED | SGD | $38.20_{\pm0.57}$ | $2.47_{\pm0.02}$ | $33.77_{\pm0.59}$ | $2.71_{\pm0.03}$ | $31.70_{\pm0.75}$ | $2.83_{\pm0.03}$ | $28.56_{\pm0.77}$ | $\mathbf{3.03}_{\pm0.04}$ | $23.72_{\pm0.78}$ | $3.33_{\pm0.05}$ |
| E-MCMC | SGLD | $36.49_{\pm0.89}$ | $2.57_{\pm0.06}$ | $32.25_{\pm0.76}$ | $2.83_{\pm0.06}$ | $30.22_{\pm0.63}$ | $2.97_{\pm0.05}$ | $27.17_{\pm0.38}$ | $3.19_{\pm0.03}$ | $22.54_{\pm0.27}$ | $3.54_{\pm0.01}$ |
| PTL | SGLD | $36.43_{\pm0.35}$ | $2.53_{\pm0.03}$ | $32.24_{\pm0.40}$ | $2.76_{\pm0.03}$ | $30.20_{\pm0.42}$ | $2.87_{\pm0.03}$ | $27.17_{\pm0.55}$ | $3.06_{\pm0.04}$ | $22.56_{\pm0.54}$ | $3.36_{\pm0.05}$ |
| SA-BMA (VI) | | $\mathbf{39.41}_{\pm0.72}$ | $\mathbf{2.44}_{\pm0.04}$ | $\mathbf{35.07}_{\pm0.64}$ | $2.70_{\pm0.05}$ | $\mathbf{32.75}_{\pm0.71}$ | $2.86_{\pm0.05}$ | $\mathbf{29.41}_{\pm0.67}$ | $3.10_{\pm0.05}$ | $\mathbf{24.25}_{\pm0.70}$ | $3.44_{\pm0.05}$ |

(c) VIT-B/16 CIFAR10C

| Method | Optim | Severity | | | | | | | | | |
| | | 1 | | 2 | | 3 | | 4 | | 5 | |
| | | ACC ↑ | NLL ↓ | ACC ↑ | NLL ↓ | ACC ↑ | NLL ↓ | ACC ↑ | NLL ↓ | ACC ↑ | NLL ↓ |
| DNN | SGD | $79.62_{\pm0.56}$ | $0.64_{\pm0.06}$ | $76.47_{\pm0.67}$ | $0.73_{\pm0.06}$ | $74.10_{\pm0.83}$ | $0.79_{\pm0.05}$ | $70.42_{\pm1.23}$ | $0.90_{\pm0.05}$ | $64.41_{\pm1.85}$ | $1.08_{\pm0.05}$ |
| | SAM | $79.78_{\pm0.49}$ | $0.61_{\pm0.01}$ | $76.59_{\pm0.64}$ | $0.70_{\pm0.02}$ | $74.58_{\pm0.94}$ | $0.75_{\pm0.02}$ | $71.12_{\pm1.06}$ | $0.86_{\pm0.03}$ | $65.26_{\pm1.46}$ | $1.03_{\pm0.04}$ |
| | FSAM | $79.87_{\pm0.83}$ | $0.62_{\pm0.02}$ | $76.78_{\pm0.78}$ | $0.70_{\pm0.02}$ | $74.70_{\pm0.60}$ | $0.76_{\pm0.01}$ | $71.29_{\pm0.49}$ | $0.86_{\pm0.01}$ | $65.53_{\pm0.56}$ | $1.03_{\pm0.03}$ |
| SWAG | SGD | $76.58_{\pm1.69}$ | $1.21_{\pm0.04}$ | $73.45_{\pm1.98}$ | $1.25_{\pm0.04}$ | $71.20_{\pm2.18}$ | $1.29_{\pm0.04}$ | $67.54_{\pm2.46}$ | $1.35_{\pm0.04}$ | $61.65_{\pm2.82}$ | $1.44_{\pm0.04}$ |
| F-SWAG | SAM | $81.03_{\pm2.20}$ | $0.60_{\pm0.05}$ | $77.73_{\pm2.63}$ | $0.69_{\pm0.06}$ | $75.45_{\pm2.96}$ | $0.76_{\pm0.07}$ | $71.82_{\pm3.31}$ | $0.87_{\pm0.08}$ | $66.05_{\pm3.59}$ | $1.03_{\pm0.10}$ |
| VI | bSAM | $78.80_{\pm1.18}$ | $0.64_{\pm0.04}$ | $75.43_{\pm1.14}$ | $0.74_{\pm0.04}$ | $73.45_{\pm1.43}$ | $0.80_{\pm0.04}$ | $70.07_{\pm1.50}$ | $0.91_{\pm0.06}$ | $64.21_{\pm1.57}$ | $1.09_{\pm0.05}$ |
| E-MCMC | SGLD | $78.91_{\pm2.31}$ | $0.65_{\pm0.08}$ | $75.78_{\pm2.36}$ | $0.74_{\pm0.08}$ | $73.94_{\pm2.56}$ | $0.79_{\pm0.09}$ | $70.66_{\pm2.63}$ | $0.89_{\pm0.10}$ | $65.07_{\pm2.77}$ | $1.06_{\pm0.11}$ |
| PTL | SGLD | $76.26_{\pm2.46}$ | $0.74_{\pm0.06}$ | $72.36_{\pm2.41}$ | $0.83_{\pm0.06}$ | $69.61_{\pm2.46}$ | $0.90_{\pm0.07}$ | $65.47_{\pm2.52}$ | $1.01_{\pm0.07}$ | $59.04_{\pm2.26}$ | $1.18_{\pm0.06}$ |
| SA-BMA (VI) | | $\mathbf{82.89}_{\pm1.09}$ | $\mathbf{0.53}_{\pm0.04}$ | $\mathbf{79.68}_{\pm1.26}$ | $\mathbf{0.62}_{\pm0.04}$ | $\mathbf{77.30}_{\pm1.43}$ | $\mathbf{0.69}_{\pm0.05}$ | $\mathbf{73.41}_{\pm1.62}$ | $\mathbf{0.81}_{\pm0.06}$ | $\mathbf{66.94}_{\pm1.79}$ | $\mathbf{1.01}_{\pm0.07}$ |

(d) VIT-B/16 CIFAR100C

| Method | Optim | Severity | | | | | | | | | |
| | | 1 | | 2 | | 3 | | 4 | | 5 | |
| | | ACC ↑ | NLL ↓ | ACC ↑ | NLL ↓ | ACC ↑ | NLL ↓ | ACC ↑ | NLL ↓ | ACC ↑ | NLL ↓ |
| DNN | SGD | $62.19_{\pm0.52}$ | $1.42_{\pm0.02}$ | $57.81_{\pm0.37}$ | $1.61_{\pm0.02}$ | $55.04_{\pm0.14}$ | $1.73_{\pm0.02}$ | $50.73_{\pm0.24}$ | $1.93_{\pm0.01}$ | $44.12_{\pm0.39}$ | $2.24_{\pm0.01}$ |
| | SAM | $61.90_{\pm0.53}$ | $1.47_{\pm0.02}$ | $57.49_{\pm0.43}$ | $1.65_{\pm0.02}$ | $54.80_{\pm0.29}$ | $1.76_{\pm0.01}$ | $50.52_{\pm0.25}$ | $1.96_{\pm0.01}$ | $44.04_{\pm0.24}$ | $2.26_{\pm0.01}$ |
| | FSAM | $61.70_{\pm0.52}$ | $1.47_{\pm0.02}$ | $57.16_{\pm0.44}$ | $1.65_{\pm0.02}$ | $54.46_{\pm0.37}$ | $1.77_{\pm0.02}$ | $50.11_{\pm0.39}$ | $1.97_{\pm0.01}$ | $43.53_{\pm0.42}$ | $2.28_{\pm0.01}$ |
| SWAG | SGD | $59.19_{\pm0.90}$ | $2.00_{\pm0.03}$ | $55.45_{\pm0.88}$ | $2.12_{\pm0.03}$ | $53.34_{\pm0.94}$ | $2.19_{\pm0.03}$ | $49.44_{\pm0.81}$ | $2.33_{\pm0.03}$ | $43.71_{\pm0.93}$ | $2.53_{\pm0.03}$ |
| F-SWAG | SAM | $59.55_{\pm2.94}$ | $1.49_{\pm0.11}$ | $55.10_{\pm2.82}$ | $1.70_{\pm0.10}$ | $52.37_{\pm2.80}$ | $1.82_{\pm0.10}$ | $48.18_{\pm2.63}$ | $2.04_{\pm0.09}$ | $41.84_{\pm2.43}$ | $2.37_{\pm0.09}$ |
| VI | bSAM | $62.36_{\pm0.73}$ | $1.40_{\pm0.03}$ | $57.97_{\pm0.70}$ | $1.58_{\pm0.03}$ | $55.32_{\pm0.61}$ | $1.70_{\pm0.03}$ | $51.09_{\pm0.49}$ | $1.90_{\pm0.02}$ | $44.77_{\pm0.42}$ | $2.21_{\pm0.03}$ |
| E-MCMC | SGLD | $62.28_{\pm0.47}$ | $1.40_{\pm0.02}$ | $57.84_{\pm0.46}$ | $1.59_{\pm0.02}$ | $55.14_{\pm0.29}$ | $1.71_{\pm0.02}$ | $50.87_{\pm0.21}$ | $1.91_{\pm0.02}$ | $44.49_{\pm0.13}$ | $2.22_{\pm0.02}$ |
| PTL | SGLD | $61.84_{\pm0.33}$ | $1.47_{\pm0.02}$ | $57.36_{\pm0.22}$ | $1.66_{\pm0.02}$ | $54.47_{\pm0.08}$ | $1.78_{\pm0.01}$ | $50.03_{\pm0.23}$ | $1.98_{\pm0.01}$ | $43.34_{\pm0.36}$ | $2.29_{\pm0.01}$ |
| SA-BMA (VI) | | $\mathbf{63.91}_{\pm0.02}$ | $\mathbf{1.33}_{\pm0.00}$ | $\mathbf{59.70}_{\pm0.00}$ | $\mathbf{1.51}_{\pm0.00}$ | $\mathbf{57.00}_{\pm0.01}$ | $\mathbf{1.63}_{\pm0.00}$ | $\mathbf{52.51}_{\pm0.03}$ | $\mathbf{1.84}_{\pm0.00}$ | $\mathbf{45.39}_{\pm0.04}$ | $\mathbf{2.18}_{\pm0.00}$ |

# H ADDITIONAL LOSS SURFACE OF SAMPLED MODEL

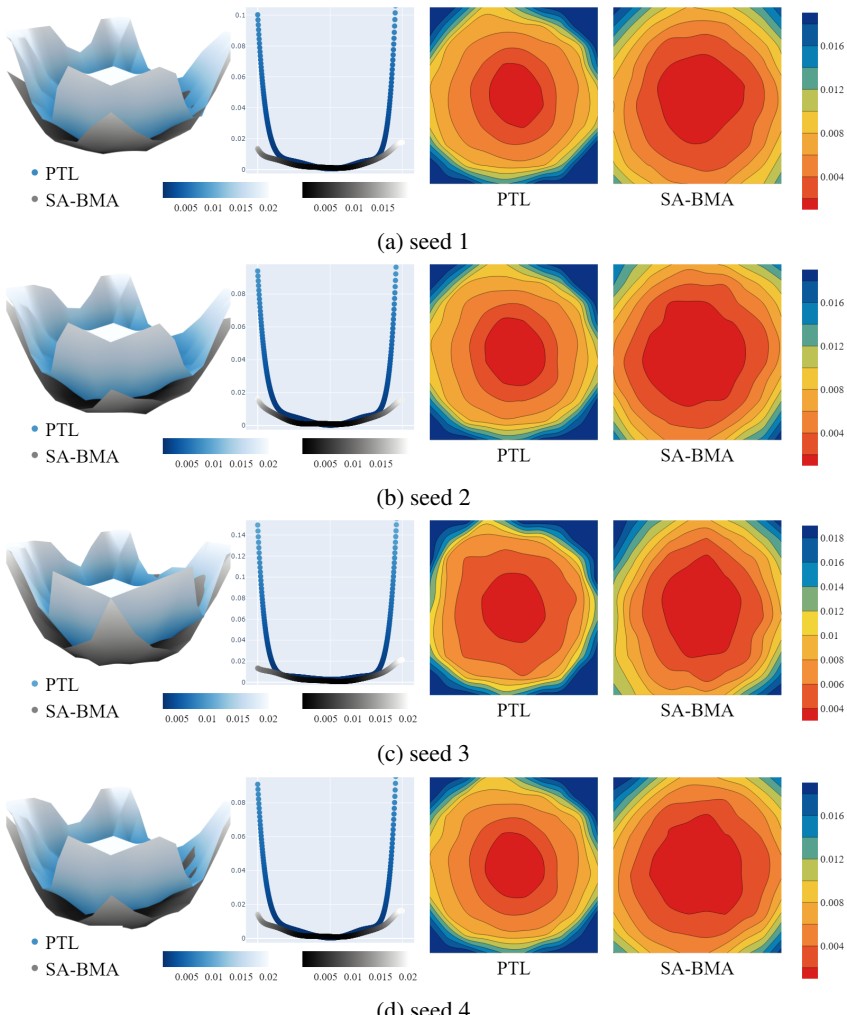

Figure 15: Four instances of sampled weights, including (b) as presented in Figure 5. Across all plots, it is consistently observed that SA-BMA converges to a flatter loss surface compared to PTL.

As shown in Figure 5, we sampled four model parameters from the posterior, which were trained on CIFAR10 with RN18. It shows the consistent and robust trend of flatness of SA-BMA in the loss surface. In Figure 15, commencing with the leftmost panel, a 3D surface plot illustrates the loss surface, revealing the SA-BMA model's comparatively flatter topology against the PTL model. This initial plot intuitively demonstrates that the SA-BMA model exhibits a flatter loss surface compared to the PTL model. Following this, the second visualization compresses the information along a diagonal plane into a 1D scatter plot. This transformation reveals areas obscured in the 3D view, highlighting that SA-BMA maintains a considerably flatter and lower-loss landscape. The third and fourth images showcase the loss surface through 2D contour plots, from which one can easily discern that the area representing the lowest loss is significantly more expansive for SA-BMA than for PTL.

# I  LIMITATION AND FUTURE WORKS

This study has several limitations. Firstly, calculating the FIM in weight space rather than output space makes it intractable to compute the FIM for the entire model parameter. Obtaining the FIM for the entire model weight space could lead to much more powerful performance improvements, making it one potential direction for future work. Secondly, the assumption of the existence of pre-trained models is necessary for Bayesian transfer learning. In situations where pre-trained weights are not available, performance improvements may be somewhat limited.

