# OpenReview forum: "Flat Posterior Does Matter For Bayesian Model Averaging"
_ICLR.cc/2025/Conference — ICLR 2025 Conference Withdrawn Submission_

### Official Review · Reviewer_ZSvu · 2024-11-02

**Soundness:** 3
**Presentation:** 3
**Contribution:** 1
**Rating:** 3
**Confidence:** 3

**Summary:**

This paper proposes to use sharpness-aware minimization for Bayesian models. It proposes a framework that can be used with 3 different posterior approximation methods.

**Strengths:**

The paper does propose an interesting combination of lines of work in deep learning, it missed out on evaluating whether this combination is useful in my opinion, though. I do see the plots in Figure 2 as a negative result in this way, and think based on this one could have written an interesting paper on flatness-seeking methods approximating Bayesian averages.

**Weaknesses:**

- I do not think there is a need for flatness-aware optimization in Bayesian models. That is because Bayesian models are building an average over all models with high likelihood (or posterior likelihood for informative priors). Taking this average will naturally lead to including a lot of models from flat optima, as they are simply wider and thus have more mass (in the prior). This in my opinion is underlined by the experiments in Figure 2b-c, where we can see that by simply using a larger Ensemble, thus approximating the true PPD more closely, we get the same effect as when choosing models that lie in wide optima. I hope I did understand this experiment right and the 30 models that you speak about are 30 samples from your posterior.
    - One more point on this: I can imagine that this argument does not work well for the particular problems that VI has, as it will always try to find a simple Gaussian distribution that represents the posterior.
- The flatness difference in Experiment 5.1 looks marginal at mere 2x radius of the optimum and a worse likelihood. This toy experiment would be more interesting if both optima had the same likelihood, but one being *much* more narrow.
- Your language is missing a lot of articles, but generally feels more like a draft than a paper. I guess you are not a native English speaker, I am neither, so this does not affect the score much for me, but I can recommend you to use LLMs/DeepL to improve your English writing.
- The accuracies seen in the experiments seem to be far away from the state of the art for the models, see e.g. this torch tutorial https://pytorch.org/blog/how-to-train-state-of-the-art-models-using-torchvision-latest-primitives/
- SAM is known to increase the time each step takes, this algorithm should have the same impact. A comparison of performance over time is missing, though.

**Questions:**

- How many posterior samples have you used for Figure 2a?
- How large are your Bayesian ensembles for the final experiments, i.e. how many times do you sample from your posterior to approximate the PPD integral?

---

### Official Review · Reviewer_tDL5 · 2024-11-04

**Soundness:** 3
**Presentation:** 3
**Contribution:** 3
**Rating:** 6
**Confidence:** 4

**Summary:**

This paper proposes a flatness-aware optimizer for Bayesian model averaging, which applies to a variety of Bayesian inference algorithms. The introduced optimizer SA-BMA is generalized from the SAM optimizer. This paper also has a clear empirical proof of why flatness is important and why existing Bayesian inference methods ignore flatness.

**Strengths:**

1. I notice that this is a resubmission paper. Compared with the last version, more analysis on the flatness of the loss landscape and the relations between flatness and general performances are included. I respect the authors' efforts in studying the geometry of loss landscape.

2. The empirical analysis using Hessian eigenvalues clearly demonstrates why finding flat modes is important to the overall performance.

3. Comprehensive experiments are conducted to demonstrate the effectiveness of SA-BMA.

**Weaknesses:**

1. The experiments on real-world datasets are limited to CIFAR10/100. I expect to see results on large-scale dataset like ImageNet to show the scalability of SA-BMA.

2. Figure 5 may lead to a misunderstanding that PTL and SA-BMA change the loss surface (in the first 2 figures).

**Questions:**

Please refer to the weaknesses.

---

### Official Review · Reviewer_JtUY · 2024-11-08

**Soundness:** 2
**Presentation:** 3
**Contribution:** 2
**Rating:** 5
**Confidence:** 3

**Summary:**

Flat optima has been shown to connect with good generalization in point estimation for neural networks. The authors study flatness for Bayesian neural networks and propose a flatness-seeking optimizer, Sharpness-Aware Bayesian Model Averaging (SA-BMA), for VI. Specifically, the authors first show empirically that (1) BNN's posterior tends to land in a sharper loss region; (2) when making a prediction with MC estimation, using flat samples will result in better performance. Based on the empirical finding, the authors propose a new learning objective for VI, which also accounts for flatness as well as a Bayesian Transfer Learning scheme for efficient computation for large models. Experiment results have shown SA-BMA can improve generalization in few-shot classification and distribution shift.

**Strengths:**

- The connection between the proposed objective and existing works is well-analyzed

- Well written and easy to follow

**Weaknesses:**

- In Bayesian deep learning in the end we have a distribution, here the authors use the averaged Hessian eigenvalues of different sampled weights as the measurement of flatness. I'm not fully convinced this is a good measurement of a flatness over a distribution.

- The proposed objective is expensive to train.

**Questions:**

- SAM training is already more expensive than vanilla gradient descent, adding VI on top (now you need to sample to estimate ELBO), won't it be too expensive? This begs another question, how much improvement can be gained by using SA-BMA when compared with LA on SAM solution? I see LA implemented in your code but there are no results of LA in the paper. I would be interested to see an experiment comparing LA on SGD solution, LA on SAM solution, and SA-BMA.

- How do you ensure VI has been trained successfully? I see in multiple cases VI ends up with higher NLL and ECE than MAP, which seems strange.

---

### Official Review · Reviewer_yjnt · 2024-11-09

**Soundness:** 2
**Presentation:** 3
**Contribution:** 2
**Rating:** 3
**Confidence:** 4

**Summary:**

The paper proposes a sharpness-aware Bayesian neural network (BNNs) to ensure the found modes are flat. A new Bayesian transfer learning scheme is also developed to leverage the pre-trained deep neural networks.

**Strengths:**

- This paper targets the generalization of BNNs, which is an important problem.
- The paper provides empirical and theoretical analysis to support the need for flatness in BNNs.

**Weaknesses:**

- The overall goal of the paper is vague. As far as I understand, the proposed method increases the flatness of the variational parameter \theta, not the model parameter w. However, the literature shows flatter w leads to better generalization. The seems to be a gap. The meaning of "flatness in BNNs" is not very clear in the paper.
- Previous works have demonstrated the benefits of including flatness in BNNs, e.g. Möllenhoff & Khan, 2022, Nguyen et al., 2023, Li & Zhang, 2023. The additional insights offered by Sec 3 are unclear.
- It is unclear how Theorem 1 indicates that BNN needs flatness. This theorem basically shows the relationship between the flatness of the weight-averaged model and the flatness of individual models. It does not explain the benefits of ensuring flatness in BNNs.
- Variational inference (VI) approximates the posterior through an optimization problem. We can naively apply SAM to the objective of VI. The difference and benefit of the proposed objectives in Eq.4 and 5 over this naive version are unclear.
- In the experiment section, the proposed method is applied to VI, SWAG, and MCMC. However, it is unclear how the method is compatible with SWAG and MCMC.
- The proposed Bayesian transfer learning scheme is a straightforward application to transfer learning. The novelty of this part is low.

**Questions:**

- How to measure the flatness of a BNN, as it involves a set of NNs? Do the authors average the flatness of all samples in Fig. 1 and 2?

---

### Note · Authors · 2024-11-14

**Comment:**

We appreciate the anonymous reviewers for their constructive feedback. Unfortunately, given the short rebuttal period, we will not be able to address all of the concerns raised in the reviews. Therefore, after careful consideration, we have decided to take time to more carefully revise our work for another venue and withdraw the current submission. We appreciate the reviewers’ time and effort, again.

**Withdrawal Confirmation:**

I have read and agree with the venue's withdrawal policy on behalf of myself and my co-authors.